# Deciphering the preeclampsia-specific immune microenvironment and the role of pro-inflammatory macrophages at the maternal–fetal interface

**Haiyi Fei**[1,2,3†], **Xiaowen Lu**[1,2,3†], **Zhan Shi**[1,2,3†], **Xiu Liu**[1,2,3], **Cuiyu Yang**[1,2,3], **Xiaohong Zhu**[4], **Yuhan Lin**[1,2,3], **Ziqun Jiang**[1,2,3], **Jianmin Wang**[1,2,3], **Dong Huang**[1,2,3], **Liu Liu**[1,2,3], **Songying Zhang**[1,2,3]*, **Lingling Jiang**[1,2,3]*

[1]Assisted Reproduction Unit, Department of Obstetrics and Gynecology, Sir Run Run Shaw Hospital, School of Medicine, Zhejiang University, Hangzhou, China; [2]Zhejiang Provincial Clinical Research Center for Reproductive Health and Disease, Hangzhou, China; [3]Zhejiang Key Laboratory of Precise Protection and Promotion of Fertility, Hangzhou, China; [4]Department of Obstetrics and Gynecology, Zhejiang Xiaoshan Hospital, Hangzhou, China

**\*For correspondence:**
zhangsongying@zju.edu.cn (SZ);
linglingjiang@zju.edu.cn (LJ)

[†]These authors contributed equally to this work

**Competing interest:** The authors declare that no competing interests exist.

## eLife Assessment

This **valuable** study investigates the immune system's role in pre-eclampsia. The authors map the immune cell landscape of the human placenta and find an increase in macrophages and Th17 cells in patients with pre-eclampsia. Following mouse studies, the authors suggest that the IGF1-IGF1R pathway might play a role in how macrophages influence T cells, potentially driving the pathology of pre-eclampsia. There is **convincing** evidence in this study that will be of interest to immunologists and developmental biologists.

**Abstract** Preeclampsia (PE), a major cause of maternal and perinatal mortality with highly heterogeneous causes and symptoms, is usually complicated by gestational diabetes mellitus (GDM). However, a comprehensive understanding of the immune microenvironment in the placenta of PE and the differences between PE and GDM is still lacking. In this study, cytometry by time of flight indicated that the frequencies of memory-like Th17 cells (CD45RA$^-$CCR7$^+$IL-17A$^+$CD4$^+$), memory-like CD8$^+$ T cells (CD38$^+$CXCR3$^-$CCR7$^+$Helios$^-$CD127$^-$CD8$^+$) and pro-inflam Macs (CD206$^-$CD163$^-$CD38$^{mid}$CD107a$^{low}$CD86$^{mid}$HLA-DR$^{mid}$CD14$^+$) were increased, while the frequencies of anti-inflam Macs (CD206$^+$CD163$^-$CD86$^{mid}$CD33$^+$HLA-DR$^+$CD14$^+$) and granulocyte myeloid-derived suppressor cells (gMDSCs, CD11b$^+$CD15$^{hi}$HLA-DR$^{low}$) were decreased in the placenta of PE compared with that of normal pregnancy (NP), but not in that of GDM or GDM&PE. The pro-inflam Macs were positively correlated with memory-like Th17 cells and memory-like CD8$^+$ T cells but negatively correlated with gMDSCs. Single-cell RNA sequencing revealed that transferring the F4/80$^+$CD206$^-$ pro-inflam Macs with a Folr2$^+$Ccl7$^+$Ccl8$^+$C1qa$^+$C1qb$^+$C1qc$^+$ phenotype from the uterus of PE mice to normal pregnant mice induced the production of memory-like IL-17a$^+$Rora$^+$Il1r1$^+$TNF$^+$Cxcr6$^+$S100a4$^+$CD44$^+$ Th17 cells via IGF1–IGF1R, which contributed to the development and recurrence of PE. Pro-inflam Macs also induced the production of memory-like CD8$^+$ T cells but inhibited the production of Ly6g$^+$S100a8$^+$S100a9$^+$Retnlg$^+$Wfdc21$^+$ gMDSCs at the maternal–fetal interface, leading to PE-like symptoms in mice. In conclusion, this study revealed the PE-specific immune cell network, which was regulated by pro-inflam Macs, providing new ideas about the pathogenesis of PE.

## Introduction

In recent years, low fertility is a global problem (*Agbaglo et al., 2022*), exacerbated by pregnancy complications. Preeclampsia (PE) is a progressive systemic disease during pregnancy, with pregnancy-induced hypertension, proteinuria, and liver and kidney injury as the main diagnostic indicators (*Chappell et al., 2021*). With a global prevalence of 7–10%, PE is a major cause of maternal and perinatal mortality and morbidity, especially in low- and middle-income countries. The only specific treatment is delivery (*Chappell et al., 2021*). However, due to the variety of symptoms and the heterogeneity of the disease, the pathogenesis of PE is still unclear (*Burton et al., 2019*; *Grotegut, 2016*).

PE is often complicated by gestational diabetes mellitus (GDM), which is characterized by insulin resistance and associated with aberrant maternal immune cell adaption (*Corrêa-Silva et al., 2018*; *McElwain et al., 2021*; *McIntyre et al., 2019*). Studies have shown that GDM is an independent risk factor for PE (*Nerenberg et al., 2013*; *Ostlund et al., 2004*; *Schneider et al., 2012*). However, whether a different pathogenesis underlies between PE and GDM is still unclear.

The fetus is equivalent to a semi-homogenous graft for the mother. Therefore, maternal–fetal immune tolerance plays an important role in the maintenance of pregnancy. Currently, the pivotal mechanism underpinning the pathogenesis of PE is widely acknowledged to involve an increased frequency of pro-inflammatory M1-like maternal macrophages (*Faas et al., 2014*; *Yao et al., 2019*), along with an elevation in granulocytes that capable of superoxide generation (*Lampé et al., 2011*; *Liu et al., 2021*), CD56[+] CD94[+] natural killer (NK) cells (*Bachmayer et al., 2006*; *Travis et al., 2020*), CD19[+]CD5[+] B1 lymphocytes (*Jensen et al., 2012*; *LaMarca et al., 2011*; *Zhong et al., 2007*), and activated γδ T cells (*Chatterjee et al., 2017*). Conversely, this pathological process is accompanied by a notable decrease in the frequency of anti-inflammatory M2-like macrophages (*Faas et al., 2014*; *Yao et al., 2019*) and NKp46[+] NK cells (*Fukui et al., 2011*). Adaptive immune response is also critical for the pathogenesis of PE (*Deer et al., 2023*). In PE, CD4[+] T cells prefer to differentiate into pro-inflammatory Th1 and Th17 (*Eghbal-Fard et al., 2019*; *Fu et al., 2014*; *Lang et al., 2021*; *Lu et al., 2020*; *Saito et al., 2007*) instead of non-inflammatory Th2 and Treg (*Care et al., 2018*; *Cornelius et al., 2015a*; *Saito et al., 2007*; *Santner-Nanan et al., 2009*; *Sasaki et al., 2007*).

Moreover, PE can be subdivided into early- and late-onset PE diagnosed before 34 weeks or from 34 weeks of gestation, respectively. Research has revealed that among the myriad of cellular alterations in PE, pro-inflammatory M1-like macrophages and intrauterine B1 cells display an augmented presence at the maternal–fetal interface of both early- and late-onset PE patients. Decidual natural killer cells and neutrophils emerge as paramount contributors, playing a more crucial role in the pathogenesis of early-onset PE (*Aneman et al., 2020*). However, a comprehensive and in-depth understanding of the maternal–fetal interface of PE is still lacking.

The immune cells crosstalk with each other at the maternal–fetal interface elaborately. For example, the interaction between B2 and Th cells depends on CD40 and its ligand CD40L, which helps B2 cells provide AT1-AAs and differentiate into memory B cells in reducing uterine perfusion pressure (RUPP) rats (*Cornelius et al., 2015b*). The cytokines derived from NK cells can promote the differentiation of Th17 to co-work in recurrent miscarriage (*Fu et al., 2013*). The intricate interplay and precise coordination between immune cells are pivotal in maintaining pregnancy. Nevertheless, the regulation of the immune network at the maternal–fetal interface of PE patients still needs to be further studied.

In this study, we combined cytometry by time of flight (CyTOF), single-cell RNA sequencing (scRNA-seq), and rodent experiment to identify the overall immune cell composition and their

**Table 1.** Details of the individuals included in the cytometry by time of flight (CyTOF).

| Parameters | NP (*n* = 9) | PE (*n* = 8) | GDM (*n* = 8) | GDM&PE (*n* = 7) | p value |
|---|---|---|---|---|---|
| Age (years) | 31.33 ± 2.108 | 31.75 ± 4.206 | 35.25 ± 3.419 | 32.43 ± 2.321 | 0.1008 |
| BMI (kg/m²) | 27.35 ± 1.740 | 30.84 ± 2.888 | 29.56 ± 2.689 | 30.16 ± 2.280 | 0.0518 |
| Gestational age (weeks) | 38.89 ± 0.558 | 37.39 ± 1.253 | 38.73 ± 0.419 | 36.22 ± 1.372 | <0.001 |
| Number of living children | 0.4444 ± 0.497 | 0.500 ± 0.500 | 1.125 ± 0.331 | 0.429 ± 0.495 | **0.0236** |
| Previous abortions | 0.7778 ± 1.030 | 1.000 ± 0.866 | 1.000 ± 1.000 | 1.714 ± 2.050 | 0.5848 |
| Mean systolic blood pressure | 109.7 ± 5.400 | 151 ± 7.225 | 106.9 ± 2.848 | 150.7 ± 9.161 | <0.0001 |

interactions at the maternal–fetal interface in PE, GDM, and GDM complicated with PE (GDM&PE). This study revealed the PE-specific immune cell network, which was regulated by pro-inflammatory macrophages, providing new ideas about the pathogenesis of PE.

## Results

### Overall immune cell profile in the placenta of PE, GDM, and GDM&PE

To fully characterize the immune microenvironment at the maternal–fetal interface of PE, we collected full-term placentas of PE and NP, and did a CyTOF test (*Table 1*), which panel is shown in *Table 2*. Since PE is usually complicated by GDM, the subjects of GDM and GDM complicated with PE (GDM&PE) were also included. The experimental workflow for CyTOF is shown in the schematic diagram (*Figure 1A*). We found that in addition to the well-known macrophages and T cells, there are also γδ T cells, B cells, NK cells, granulocytes, dendritic cells, and myeloid-derived suppressor cells (MDSCs). An overall distribution of CD45$^+$ cell subsets in placenta as shown in the t-Distributed Stochastic Neighbor Embedding (t-SNE) maps (*Figure 1B*). There were no significant differences in the proportion of these large subpopulations comparing the PE, GDM, GDM&PE group to the NP group (*Figure 1C*). To identify the 10 cell subpopulations, CD4, CD8, and γδTCR were used as markers to distinguish CD4$^+$ T, CD8$^+$ T, and γδ T cells. CD11b is a marker that distinguishes myeloid cells, including macrophages and granulocytes (*Figure 1D*). The 10 cell subsets were further annotated by 41 markers in the heatmap (*Figure 1E*, *Table 3*).

### Specific altered T cell profile in the placenta of PE

To gain a comprehensive understanding of the distribution patterns within each cellular subset, we conducted separate clustering analyses for CD4$^+$ T, CD8$^+$ T, and γδ T cells, respectively. We first analyzed 15 clusters of CD4$^+$ T cells from the placentas of NP, PE, GDM, and GDM&PE (*Figure 2A*). CD4$^+$ T cell clusters were defined by canonical marker set signatures (*Figure 2—figure supplement 1A*). The frequencies of memory-like CD45RA$^-$CCR7$^+$CD69$^-$CD127$^{low}$TIGIT$^-$Helios$^-$CD4$^+$ T cells (cluster 8) were significantly increased in individuals with PE compared with NP controls. At the same time, there was no significance in the GDM and GDM&PE groups compared with the NP group (*Figure 2B*, *Table 4*). We analyzed the expression of common intracellular molecules in CD4$^+$ memory-like T cells by sorting CD45RO$^+$CCR7$^+$CD4$^+$ T cells from placental samples by flow cytometry. Higher levels of IL-17A and lower levels of Foxp3 were found in memory-like CD45RO$^+$CCR7$^+$CD4$^+$ T cells in the placentas of individuals with PE (*Figure 2C*). Consistent with the results of CyTOF, higher fluorescence intensity of CD45RO$^+$CD4$^+$ T cells was found in the placentas of individuals with PE than that of NP, and this cell population was mainly located in the placental sinusoids (*Figure 2D*), suggesting that these T cells should be recently recruited. Moreover, lower expression of immune checkpoint molecules including T-cell immunoglobulin mucin-3 (Tim-3) and programmed cell death 1 (PD-1) was found in CD45RA$^-$CCR7$^+$CD4$^+$ memory T cells in the PE group, suggesting that these cells have a lower immunosuppressive capacity (*Chen et al., 2022b*; *Fanelli et al., 2021*; *Rasmussen et al., 2022*; *Wang et al., 2016*), there was no significant difference in the GDM or GDM&PE groups comparing with the NP group (*Figure 2E*). Then, 18 clusters of CD8$^+$ T cells were analyzed (*Figure 2F*). CD8$^+$ T cell clusters annotation was based on the expression of canonical marker signatures (*Figure 2—figure supplement 1B*). CD38$^+$CXCR3$^-$CCR7$^+$Helios$^-$CD127$^-$ (cluster 2) memory-like CD8$^+$ T cells were significantly increased in the PE group (*Figure 2G*, *Table 4*). CD38$^+$CXCR3$^-$CCR7$^+$Helios$^-$CD127$^-$ memory-like CD8$^+$ T cells of the PE group expressed lower levels of PD-1 and TIGIT, suggesting the activation of cytotoxicity of these cells (*Figure 2H*; *Morita et al., 2020*). Then, 22 clusters of γδT cells were analyzed and annotated (*Figure 2—figure supplement 1C, D*). Cluster 15 (Helios$^{mid}$CD28$^{mid}$CD69$^{mid}$HLA-DR$^{mid}$CD127$^{mid}$γδT cells), with lower expression of Tim3, was significantly decreased in the PE group (*Figure 2—figure supplement 1E, F*). Nevertheless, neither the CD4$^+$ T nor the CD8$^+$ T cells exhibited any alteration in the memory-like clusters when comparing the GDM or GDM&PE groups to the NP group.

In conclusion, significant changes in placental T cell profile were found in the placentas of PE but not in those of GDM or GDM&PE, suggesting that abnormal activation of T cells in the placenta is associated with the pathogenesis of PE.

**Table 2.** Cytometry by time of flight (CyTOF) antibody panel used for analyzing placentas from individuals with normal pregnancy (NP), preeclampsia (PE), gestational diabetes mellitus (GDM), and GDM complicated with PE (GDM&PE).

| Target | Metal tag |
| --- | --- |
| CD45 | HI30 |
| CD3 | UCHT1 |
| CD68 | Y1/82A |
| CD56 | NCAM16.2 |
| gd TCR | 5A6.E9 |
| CD19 | HIB19 |
| CCR6 | G034E3 |
| CD38 | HIT2 |
| CD103 | B-Ly7 |
| CD39 | A1 |
| CXCR3 | G025H7 |
| PD-L1 | 29E.2A3 |
| PD-1 | EH12.2H7 |
| CD11C | BU15 |
| CD107a | H4A3 |
| pAKT | D9E |
| CCR4 | L291H4 |
| TIGIT | A15153G |
| CD206 | 15-2 |
| Helios | 22F6 |
| CD28 | CD28.2 |
| pSTAT3 | 4/p-Stat3 |
| GITR | 110416 |
| CD33 | WM53 |
| CTLA-4 | BN13 |
| FOXP3 | PCH101 |
| CD163 | GHI/61 |
| CD45RA | HI100 |
| ICOS | C398.4A |
| CD69 | FN50 |
| CCR7 | G043H7 |
| NKG2A | 131411 |
| CD15 | W6D3 |
| TIM-3 | F38-2E2 |
| CD86 | Fun-1 |
| HLA-DR | L243 |
| Granzyme B | QA16A02 |
| CD14 | M5E2 |

*Table 2 continued on next page*

*Table 2 continued*

| Target | Metal tag |
| --- | --- |
| pS6 | A17020B |
| CD127 | A019D5 |
| CD4 | RPA-T4 |
| CD8 | RPA-T8 |
| CD11b | M1/70 |

## Abnormal polarization of macrophages was correlated with specific immune cell subsets in individuals with PE

Except for T cells, we also analyzed 29 clusters of CD45+CD3−CD11b+ cells from placentas of NP, PE, GDM, and GDM&PE, including 9 clusters of macrophages, 11 clusters of granulocytes, and 5 clusters of NK/NK-like cells (*Figure 3A*). The clusters of CD45+CD3−CD11b+ cells were defined by canonical marker set signatures (*Figure 3B*). Moreover, significantly decreased frequencies of CD68midCD39midHLA-DRlowCD11b+CD15+ granulocytes (cluster 12), which are identified as gMDSCs, were found in the PE group (*Figure 3C*, *Table 4*). In addition, the frequency of anti-inflammatory macrophages (anti-inflam Macs) (CD206+CD163−CD86midCD33+HLA-DR+, cluster 25) was also significantly decreased, whereas the frequency of pro-inflammatory macrophages (pro-inflam Macs) (CD206−CD163−CD38midCD107alowCD86midHLA-DRmidCD14+, cluster 23) was significantly increased in the PE group (*Figure 3C*, *Table 4*). However, the frequencies of macrophages and gMDSCs were unchanged significantly in the GDM or GDM&PE groups.

Pearson correlation analysis indicated positive correlations between pro-inflam Macs (cluster 23 in CD11b+ cells) and CD4+ memory-like T cells (cluster 8 of CD4+ T cells), as well as CD8+ memory-like T cells (cluster 2 in CD8+ T cells). CD206− pro-inflam Macs were negatively correlated with gMDSCs (cluster 12 in CD11b+ cells), but were not statistically significant. Conversely, CD206+ anti-inflam Macs (cluster 25 in CD11b+ cells) were positively correlated with gMDSCs and negatively correlated with CD8+ memory-like T cells (*Figure 3D, E*).

These results suggested that abnormally polarized CD206− pro-inflam Macs are positively correlated with memory-like CD4+ and CD8+ T cells in the placentas of individuals with PE and negatively correlated with gMDSCs.

## F4/80+CD206− pro-inflam Macs induced immune imbalance at the maternal–fetal interface and PE-like symptoms

Though increased pro-inflam Macs have been reported in the placenta from individuals with PE (*Faas et al., 2014*), few studies have reported the interaction between macrophages and other immune cells in the placenta. In this study, transcriptome RNA-seq was used to analyze the difference between CD45+F4/80+CD206− pro-inflam Macs and CD45+F4/80+CD206+ anti-inflam Mac isolated from the uterus and placentas of mice with reducing uterine perfusion pressure (RUPP) (*Figure 4A*). The diagrammatic representation of the RUPP model is presented in *Figure 4—figure supplement 1A*. To establish the PE mouse model, we ligated uterine arteries in pregnant mice on day 12.5 of gestation, and mice with sham operation were considered as controls (*Figure 4—figure supplement 1A*). We found that mice of RUPP showed an increased embryo absorption rate, decreased fetal weight, and pup crown-rump length compared with the sham operation group (*Figure 4—figure supplement 1B*). Increased systolic blood pressure (SBP) and urine albumin creatine ratio (UACR) were also observed in mice with RUPP, which indicated a successful PE mice mode was built (*Figure 4—figure supplement 1C*). Then, we injected the PE mice derived macrophages into normal pregnant mice. An increased embryo resorption rate, decreased fetal top-rump length and fetal weight were found in mice injected with PE mice derived macrophages compared with mice injected PBS (*Figure 4—figure supplement 1D*). Increased SBP and UACR were also found in mice injected with PE mice derived macrophages (*Figure 4—figure supplement 1E*).

In addition, the green fluorescent protein (GFP) pregnant mice were used to distinguish between maternal- and fetal-derived macrophages. The wild-type (WT) female mice were mated with either

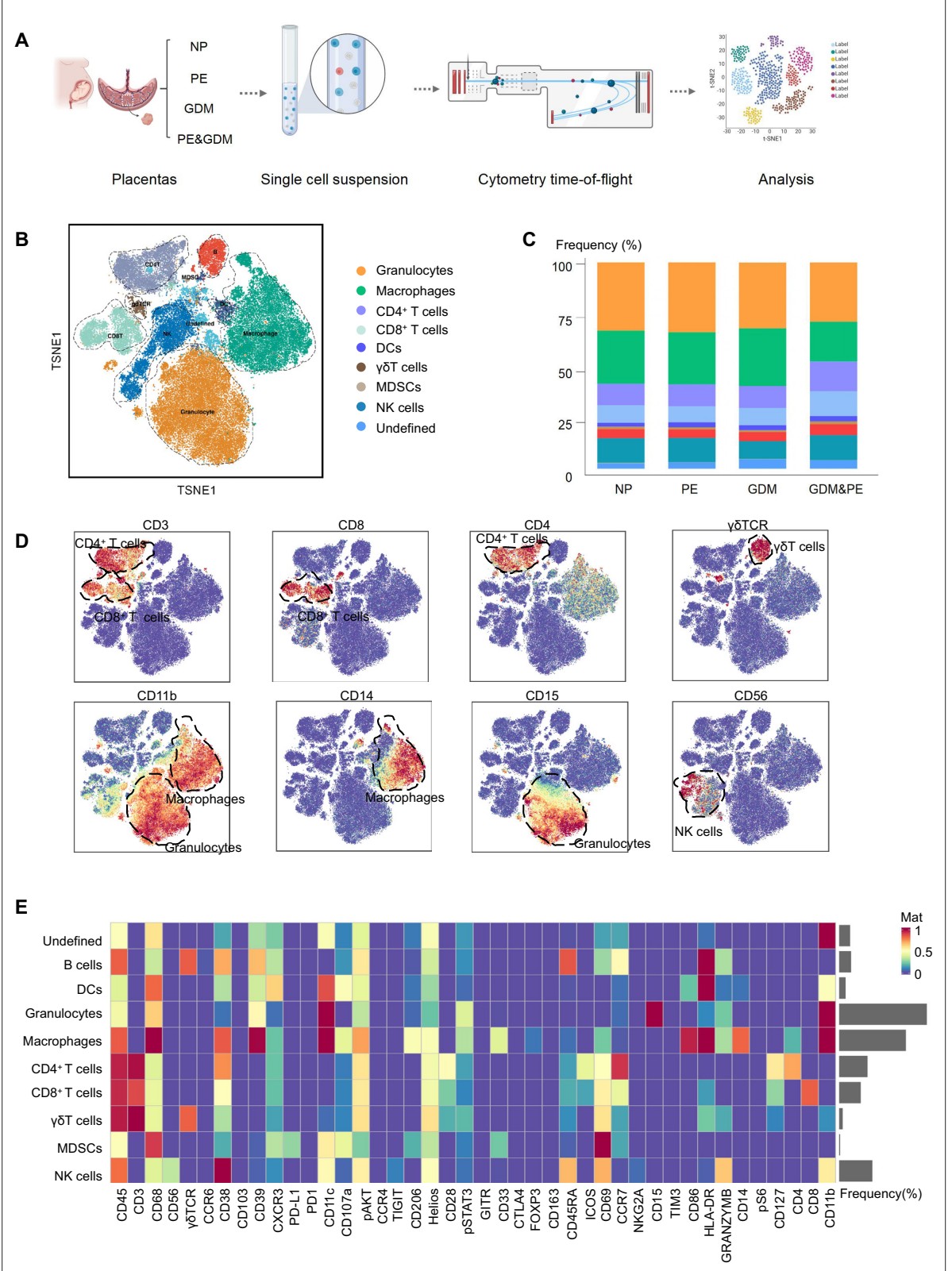

**Figure 1.** Identification and characterization of placental immune cells using cytometry by time of flight (CyTOF). (**A**) Schematic of the experimental workflow in CyTOF experiment. The placentas were obtained from individuals with normal pregnancy (NP, *n* = 9), preeclampsia (PE, *n* = 8), gestational diabetes mellitus (GDM, *n* = 8), or GDM&PE (*n* = 7). (**B**) t-Distributed Stochastic Neighbor Embedding (t-SNE) maps showing 8 × 10⁴ CD45⁺ cells (the average cell number of the all samples) from the placenta overlaid with color-coded clusters and the distributions of B cells, CD4⁺ T cells, CD8⁺ T cells,

*Figure 1 continued on next page*

*Figure 1 continued*

dendritic cell (DC), γδT cells, monocytes, granulocytes, myeloid-derived suppressor cell (MDSC), and natural killer (NK) cells. (**C**) Percentages of each cell type of CD45$^+$ cells in placentas. (**D**) t-SNE maps showing the expression of CD3, CD8, CD4, γδTCR, CD14, CD15, and CD56. (**E**) Heatmap showing the expression levels of markers in CD45$^+$ cell subsets. Data were compared between NP and PE, NP and GDM, and NP and GDM&PE using the Kruskal–Wallis test and represented as mean ± SEM (*$p < 0.05$, NS, not significant). CyTOF soure data are stored in Dryad Digital Repository, DOI: 10.5061/dryad.4qrfj6qn0.

The online version of this article includes the following source data for figure 1:

**Source data 1.** Raw data and detailed analysis for *Figure 1*.

transgenic male mice, genetically modified to express GFP to generate either GFP-expressing pups (GFP$^+$pups), or mated with WT male mice for control (*Figure 4—figure supplement 1F*). We found that the majority of macrophages in the uterus and placenta are of maternal origin (CD11b$^+$GFP$^-$). In contrast, fetal-derived macrophages (CD11b$^+$GFP$^+$) represent a mere fraction of the total macrophage population (*Figure 4—figure supplement 1G*). In light of these findings, we have incorporated mouse uterine tissues to isolate macrophages for subsequent experiments with enhanced effectiveness.

Significant differences were found between the pro-inflam and anti-inflam macrophages derived from RUPP mouse (*Figure 4B*). There are 2123 genes increased, and 2699 genes decreased significantly in the pro-inflam Macs compared to the anti-inflam Macs (*Figure 4C*). The expression levels of genes associated with inflammatory response (Ccl7, Ccl8, Ccl2, and IL6) (*He et al., 2019*; *Wu et al., 2023*), complement system activation (C1qa and C1qb) (*Chen et al., 2021*), and lipid metabolism (Pltp and Apoe) (*Desrumaux et al., 2016*) were significantly increased in the pro-inflam Macs; whereas the expression levels of genes associated with the regulation of angiogenesis and vascular endothelial growth factor production (VEGFα and Macro) and tissue development (TGF-β1, Thbs1, Fn1, and Slpi) (*Jin et al., 2023*; *Nugteren and Samsom, 2021*; *Li et al., 2022a*) were significantly decreased (*Figure 4C, D*). Moreover, gene set enrichment analysis was performed to explore the most significantly enriched functional terms between the two groups of macrophages. We found that NF-κB signaling and interspecies interaction between organisms were enriched in the pro-inflam Macs, while tissue development and epithelial cell morphogenesis were enriched in the anti-inflam Macs (*Figure 4E*).

To further investigate the effect of the pro-inflam Macs in immune imbalance at the maternal–fetal interface, PLX3397, the inhibitor of CSF1R, which is needed for macrophage development, was used to deplete the macrophages of pregnant mice (*Chen et al., 2023*; *Chen et al., 2022a*; *Figure 4—figure supplement 2A*). As expected, an increased embryo resorption rate, decreased fetal top-rump length and fetal weight were found in mice injected with pro-inflam Macs (*Figure 4F*). Increased SBP and UACR were also found in mice injected with pro-inflam Macs (*Figure 4G*). Moreover, it is shown that CD44$^+$ memory-like Th17 cells and memory-like CD8$^+$ T cells increased while CD11b$^+$Ly6G$^+$ gMDSCs decreased in mice injected pro-inflam Macs compared with mice injected anti-inflam Macs, which validated our findings with CyTOF (*Figure 4H*). Clodronate liposomes were also used to deplete the macrophages of pregnant mice (*Liu et al., 2022*) before pro-inflam or anti-inflam Macs were injected into the mice. The same experimental results were obtained (*Figure 4—figure supplement 2B–2E*). In conclusion, the F4/80$^+$CD206$^-$ pro-inflam Macs induced immune imbalance at the maternal–fetal interface and PE-like symptoms.

## Pro-inflam and anti-inflam Macs subsets are phenotypically heterogeneous

To further explore the role that macrophages play in the immune imbalance at the maternal–fetal interface in PE, scRNA-seq was performed to analyze the CD45$^+$ immune cells in the uterus and placenta from mice that were injected with pro-inflam or anti-inflam Macs from the uterus and placentas of RUPP mice. An unsupervised cluster detection algorithm (SEURAT) was applied and eight types of immune cells were detected by mostly distinguishable cell type-specific genes, including macrophages, monocytes, granulocytes, T/NK cells, B cells, DC-like cells, mast cells, and basophils (*Figure 5A, B*). Macrophages were further identified into 15 clusters defined by marker set signatures. Single-cell differential expression analysis was performed for each population and characteristic gene expression patterns were detected for clusters 0–14 to characterize the phenotypes of these subsets in detail (*Figure 5C*, *Figure 5—figure supplement 1C*). We found that the frequency of cluster 0

**Table 3.** Expression levels of markers identified in each immune subset in the placentas.

| | B cells | CD4+ T | CD8+ T | gdTCR | NK cells | Macrophage | Granulocyte | MDSC | Dendritic cells |
|---|---|---|---|---|---|---|---|---|---|
| CD45 | ++ | ++ | ++ | ++ | ++ | ++ | ++ | ++ | ++ |
| CD3 | − | ++ | ++ | − | − | − | − | − | − |
| CD68 | + | + | + | + | + | ++ | + | ++ | ++ |
| CD56 | − | − | − | − | + | − | − | − | − |
| gd TCR | ++ | − | − | ++ | − | − | − | − | − |
| CD19 | ++ | − | − | ++ | − | − | − | − | − |
| CCR6 | + | − | − | − | − | − | − | − | − |
| CD38 | ++ | ++ | + | + | ++ | ++ | − | − | + |
| CD103 | − | − | − | − | − | − | − | − | − |
| CD39 | ++ | − | − | − | + | ++ | + | + | + |
| CXCR3 | + | + | + | + | + | + | − | + | ++ |
| PD-L1 | − | − | − | − | − | − | − | + | − |
| PD-1 | − | − | − | − | − | − | − | − | − |
| CD11C | + | − | − | − | + | ++ | ++ | + | ++ |
| CD107a | + | − | − | − | − | + | − | + | + |
| pAKT | + | + | + | + | + | ++ | + | + | + |
| CCR4 | − | − | − | − | − | − | − | − | − |
| TIGIT | − | − | − | − | − | − | − | − | − |
| CD206 | − | − | − | − | − | + | − | + | − |
| Helios | + | + | + | + | + | + | + | + | + |
| CD28 | − | + | + | + | − | − | − | − | − |
| pSTAT3 | + | + | − | + | − | + | + | − | − |
| GITR | − | − | − | − | − | − | − | − | − |
| CD33 | − | − | − | − | − | + | − | + | − |
| CTLA-4 | − | − | − | − | − | − | − | − | − |
| FOXP3 | − | − | − | − | − | − | − | − | − |
| CD163 | − | − | − | − | − | − | − | − | − |
| CD45RA | ++ | + | + | + | ++ | − | − | − | − |
| ICOS | − | + | + | − | − | − | − | − | − |
| CD69 | + | + | + | + | ++ | + | − | ++ | − |
| CCR7 | + | ++ | + | + | − | − | − | + | − |
| NKG2A | − | − | − | − | − | − | − | − | − |
| CD15 | − | − | − | − | − | − | ++ | − | − |
| TIM-3 | − | − | − | − | − | − | − | − | − |
| CD86 | − | − | − | − | − | ++ | − | − | + |
| HLA-DR | ++ | − | + | + | − | ++ | + | − | ++ |
| Granzyme B | + | − | − | + | ++ | + | + | − | − |
| CD14 | − | − | − | − | − | ++ | − | − | − |
| pS6 | − | − | − | − | − | − | − | − | − |

*Table 3 continued on next page*

Table 3 continued

| | B cells | CD4+ T | CD8+ T | gdTCR | NK cells | Macrophage | Granulocyte | MDSC | Dendritic cells |
|---|---|---|---|---|---|---|---|---|---|
| CD127 | − | ++ | + | − | − | − | − | − | − |
| CD4 | − | ++ | − | − | − | − | − | − | − |
| CD8 | − | − | ++ | − | − | − | − | − | − |
| CD11b | − | − | − | + | + | ++ | ++ | − | + |

was significantly increased in mice injected with pro-inflam Macs, while the frequency of cluster 1 was significantly decreased (*Figure 5D*). We also found that cluster 0 highly expresses genes associated with fetal and tissue resident (Folr2) (*Nalio Ramos et al., 2022*; *Thomas et al., 2021*), complement system activation (C1qa, C1qb, and C1qc) (*Chen et al., 2021*), and inflammatory response (Ccl7 and Ccl8) (*He et al., 2019*; *Wu et al., 2023*); while cluster 1 highly expresses genes associated with tissue repair (Chil3, Slpi, and Fn1) (*Jin et al., 2023*; *Nugteren and Samsom, 2021*; *Li et al., 2022b*), blood vessel morphogenesis (Thbs1) (*Che et al., 2021*), and preventing oxidative stress (Gsr and Mgst1) (*Coppo et al., 2022*; *Figure 5E*). Changes in gene expression patterns of cluster 0 (pro-inflam Macs) and cluster 1 (anti-inflam Macs) were analyzed in mice injected the pro-inflam or anti-inflam Macs. We found that enriched GO terms in cluster 0 and cluster1 including 'antigen processing and presentation of exogenous antigen' and 'inflammatory response' in the pro-inflam Macs group (*Figure 5F*), in which high expression of genes such as CCL5, CCL8, CALR, IRF7, IL10, IFI44, IFI30, and OAS3 could be observed (*Figure 5G*). The control group showed elevated expression of MMP14, Chil3, EGR1, ATF3, and VASP (*Figure 5G*); and these genes were enriched in 'vascular endothelial growth factor production' and 'tissue development' (*Figure 5F*). These results were consistent with the transcriptome RNA sequencing results of macrophages.

These data suggested that the CD45+F4/80+CD206− pro-inflam Macs with a Folr2+Ccl7+Ccl8+C1qa+C1qb+C1qc+ phenotype play an important role in the development of PE.

## Pro-inflam Macs induced the memory-like Th17 cells, which was associated with the development and recurrence of PE

We also analyzed the scRNA-seq data of uterine T/NK cells from mice that were injected with or without pro-inflam Macs. T/NK cells were further identified into 12 clusters defined by marker set signatures (*Figure 6A*, *Figure 6—figure supplement 1A*). We found that the frequency of cluster 0 was increased in mice injected with pro-inflam Macs, while the frequency of clusters 1 and 2 were significantly decreased (*Figure 6B*). In addition to genes associated with Th17 cells (IL-17a, IL17f, Rora, Il1r1, and TNF) (*Hang et al., 2019*; *Leite et al., 2023*), cluster 0 also exhibits high expression of genes associated with memory phenotype (Cxcr6, S100a4, and CD44) (*Evrard et al., 2023*; *Bieberich et al., 2021*), suggesting that cluster 0 was the memory-like Th17 cells (*Figure 6C*). Cluster 1 exhibited high expression of genes associated with immunoregulation (Lef1, Tcf7, and Ccr7) (*Qiu and Du, 2022*; *Sekine et al., 2020*); Cluster 2 exhibited high expression of genes associated with immunomodulation (Gata3, GITR, and CD28), suggesting that clusters 1 and 2 were T cells with immunomodulation function (*Esensten et al., 2016*; *Pai et al., 2023*; *Figure 6—figure supplement 1B*). Changes in gene expression patterns of cluster 0 (memory-like Th17 cells) were analyzed in the pro-inflam Macs and control groups. We found that in the pro-inflam Macs group, enriched GO terms including 'positive regulation of response to external stimulus' and 'tumor necrosis factor production' (*Figure 6D*), in which highly express genes such as TNFSF11, TNF, IL27RA, IGF1R, CD226, and LAMP1 (*Figure 6E*). The control group showed elevated expression of ASS1, EIF5, S100A9, CTLA4, S100A8, and CXCR4 (*Figure 6E*); these genes were enriched in GO terms including 'regulation of programmed cell death' and 'cellular biosynthetic process' (*Figure 6D*). In summary, the pro-inflam Macs induced memory-like Th17 cells, which were characterized by IL-17a+IL17f+Rora+Il1r1+TNF+Cxcr6+S100a4+CD44+.

To confirm the importance of memory-like CD4+ T cells in the pathogenesis of PE, the CD4+CD44+ T cells in the uterus and placentas from RUPP or NP mice were sorted and intravenously injected into normal pregnant mice on day 12.5 of gestation (*Figure 6F*). The CD4+CD44+ T cells derived from RUPP mouse were characterized by an increased frequency of Th17 cells and a reduced frequency of

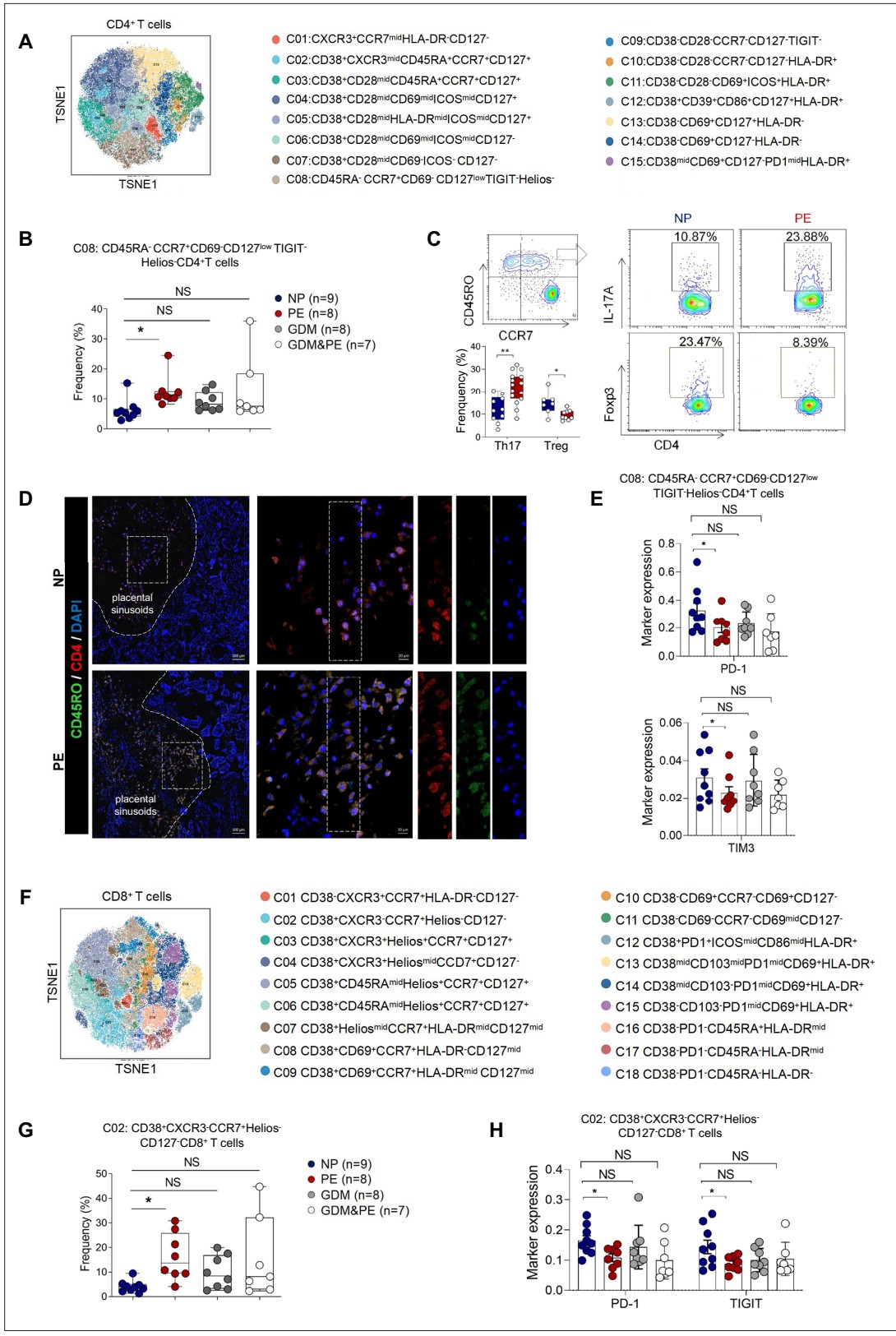

**Figure 2.** Specific altered T cell profile in the placentas of individuals with PE. (**A**) Distribution of the CD4+ T cells in the group analyzed using t-Distributed Stochastic Neighbor Embedding (t-SNE). (**B**) Scatter dot plots showing the frequencies of cluster 8 of CD4+ T cells in the placentas of individuals with NP, PE, GDM, and GDM&PE (*n* = 9 in NP group; *n* = 8 in PE group; *n* = 8 in GDM group; *n* = 7 in GDM&PE group). (**C**) Expression of IL-17A and Foxp3 in CD45RO+CCR7+CD4+ T cells in placentas of individuals with NP and PE using flow cytometry (IL-17A: *n* = 10 in NP group, *n* = 15 in

*Figure 2 continued on next page*

Figure 2 continued

PE group; Foxp3: $n$ = 7 in NP group, $n$ = 9 in PE group). (**D**) Immunofluorescence co-staining of CD4 (red), CD45RO (green), and DAPI (blue) in frozen placental sections. The right panels show the fluorescence intensity of CD4 and CD45RO. Scale bar, 20 µm. (**E**) Scatter dot plots showing significantly altered markers of in cluster 8 of CD4$^+$ T cells. (**F**) Distribution of the CD8$^+$ T cells analyzed using t-SNE. (**G**) Scatter dot plots showing the frequencies of cluster 2 of CD8$^+$ T cells in the placentas of individuals with NP, PE, GDM, and GDM&PE. (**H**) Scatter dot plots showing significantly altered markers in cluster 2 of CD8$^+$ T cells. Data were compared between NP and PE, NP and GDM, and NP and GDM&PE using the Kruskal–Wallis test and represented as mean ± SEM (*p < 0.05, NS, not significant).

The online version of this article includes the following source data and figure supplement(s) for figure 2:

**Source data 1.** Raw data and detailed analysis for *Figure 2*.

**Figure supplement 1.** Identification of the placental T cell subsets.

**Figure supplement 1—source data 1.** Raw data and detailed analysis for *Figure 2—figure supplement 1*.

Tregs (*Figure 6F*). Increased embryo absorption rate, decreased fetal weight and pup crown-rump length were found in mice injected with PE mouse-derived CD4$^+$CD44$^+$ T cells (*Figure 6G*). The PE-like symptoms, including increased SBP and UACR, were also observed in those mice (*Figure 6H*).

It has been reported that sustained expansion of immunosuppressive memory Tregs during prior pregnancy is beneficial for maintaining a second pregnancy (*Rowe et al., 2012*). To verify whether memory-like Th17 cells promote the recurrence of PE, we established a second pregnant mouse model with a history of PE or NP in the first pregnancy. An increased embryo resorption rate, decreased crown-rump length, and fetal weight were found in mice with a history of PE pregnancy compared with those with a history of NP pregnancy (*Figure 6—figure supplement 1C*). Moreover, mice with a history of PE in the first pregnancy showed increased SBP and UACR during the second pregnancy (*Figure 6—figure supplement 1D*). Consistently, higher levels of IL-17A were also found in memory-like CD4$^+$ T cells in mice with a history of PE in the first pregnancy (*Figure 6—figure supplement 1E*).

The uterine granulocytes from mice that were transferred with pro-inflam Macs from PE mice were also analyzed using scRNA-seq. Granulocytes were identified into 12 clusters defined by marker set signatures (*Figure 6—figure supplement 2A, B*). We found that the frequency of cluster 3 were decreased (*Figure 6—figure supplement 2A*). And cluster 3 highly expressed gMDSCs associated genes (Ly6g, S100a8, Retnlg, and Wfdc21) (*von Wulffen et al., 2023*; *Kao et al., 2023*; *Figure 6—figure supplement 2C*). Changes in gene expression patterns of cluster 3 (gMDSCs) were analyzed in the pro-inflam Macs and the control groups. Enriched GO terms including 'tumor necrosis factor production' and 'leukocyte mediated immunity' were found in the pro-inflam Macs group (*Figure 6—figure supplement 2E*), which highly express the genes such as IRF7, EGR1, C1QA, C1QB, CCL2, CSF3R, and TNFRSF1B (*Figure 6—figure supplement 2D*). Except for gMDSCs, we also revealed a pronounced increase in the proportion of CD15$^+$CD66b$^+$ neutrophils among PE patients, implying the importance of neutrophils in the development of PE (*Akkari et al., 2024*; *Figure 6—figure supplement 2F*).

In conclusion, pro-inflam Macs could induced the differentiation of IL-17a$^+$IL17f$^+$Rora$^+$Il1r1$^+$TNF$^+$Cxcr6$^+$S100a4$^+$CD44$^+$ memory-like Th17 cells, which was associated with the development and recurrence of PE. And the pro-inflam Macs also suppressed the production of Ly6g$^+$S100a8$^+$Retnlg$^+$Wfdc21$^+$ gMDSCs.

## Pro-inflam Macs induced the production of memory-like Th17 cells via IGF1–IGF1R

Our results above showed that pro-inflam Macs induced memory-like Th17 cells in PE, however, the underlying molecular mechanisms were still unknown. CellPhoneDB analysis indicated that increased communication counts and signaling pathways numbers between macrophages and T/NK cells were observed in mice injected with pro-inflam Macs (*Figure 7—figure supplement 1A, B*). Then we identified the interacting ligand–receptor pairs between different types of immune cells and macrophages. Insulin-like growth factor 1 (IGF1) receptor (IGF1R) ligand–receptor pair was significantly enhanced between macrophages and T/NK cells in mice injected with pro-inflam Macs (*Figure 7A*). IGF1 has significant effects on immune function maintenance, and signaling through IGF1R could cause increased aerobic glycolysis, favoring Th17 cell differentiation over that of Treg cells (*Bekkering et al., 2018*; *DiToro et al., 2020*). For further demonstration, we analyzed the frequencies of IGF1$^+$CD14$^+$

**Table 4.** The marker profile of PE-specific immune subsets.

| | Memory-like CD4+ T cells | Memory-like CD8+ T cells | gMDSCs | Pro-inflammatory macrophages | Anti-inflammatory macrophages |
|---|---|---|---|---|---|
| | Cluster 08 of CD4+ T cells | Cluster 02 of CD8+ T cells | Cluster 12 of CD3−CD11b+ cells | Cluster 23 of CD3−CD11b+ cells | Cluster 25 of CD3−CD11b+ cells |
| CD45 | ++ | ++ | ++ | ++ | ++ |
| CD3 | ++ | ++ | − | − | − |
| CD68 | - | − | + | ++ | ++ |
| CD56 | − | − | − | − | − |
| gd TCR | - | − | − | − | − |
| CD19 | - | − | − | − | − |
| CCR6 | - | − | − | − | − |
| CD38 | ++ | ++ | − | + | + |
| CD103 | − | − | − | + | + |
| CD39 | + | − | + | − | − |
| CXCR3 | + | + | − | − | − |
| PD-L1 | − | − | − | − | − |
| PD-1 | − | − | − | − | − |
| CD11C | - | − | ++ | ++ | ++ |
| CD107a | - | − | − | + | + |
| pAKT | + | + | − | − | − |
| CCR4 | − | − | − | − | − |
| TIGIT | − | − | − | − | − |
| CD206 | − | − | + | − | + |
| Helios | + | + | − | − | − |
| CD28 | + | + | − | − | − |
| pSTAT3 | - | − | − | − | − |
| GITR | − | − | − | − | − |
| CD33 | − | − | − | − | + |
| CTLA-4 | − | − | − | − | − |
| FOXP3 | − | − | − | − | − |
| CD163 | − | − | − | − | − |
| CD45RA | - | − | − | − | − |
| ICOS | + | + | − | − | − |
| CD69 | + | − | − | − | − |
| CCR7 | ++ | ++ | − | − | − |
| NKG2A | − | − | − | − | − |
| CD15 | − | − | ++ | − | − |
| TIM-3 | − | − | − | − | − |
| CD86 | − | − | − | + | + |
| HLA-DR | - | − | + | + | ++ |
| Granzyme B | - | − | + | − | − |
| CD14 | − | − | − | + | + |
| pS6 | − | − | − | − | − |

*Table 4 continued on next page*

*Table 4 continued*

| | Memory-like CD4+ T cells | Memory-like CD8+ T cells | gMDSCs | Pro-inflammatory macrophages | Anti-inflammatory macrophages |
|---|---|---|---|---|---|
| | Cluster 08 of CD4+ T cells | Cluster 02 of CD8+ T cells | Cluster 12 of CD3−CD11b+ cells | Cluster 23 of CD3−CD11b+ cells | Cluster 25 of CD3−CD11b+ cells |
| CD127 | + | − | − | − | − |
| CD4 | ++ | − | − | − | − |
| CD8 | − | ++ | − | − | − |
| CD11b | − | − | ++ | ++ | ++ |

and IGF1R+CD4+ cells in placentas of individuals with PE and NP and found both of these two cells were significantly increased in the PE group (*Figure 7B*).

Trophoblast-derived extracellular vesicles from the placenta of PE (PE-EVs), which carry the fetal antigen, could induce M1-like macrophage (pro-inflammatory Macs) polarization to participate in the development of PE according to our previous study (*Liu et al., 2022*). To confirm whether pro-inflammatory Macs induced the production of memory-like Th17 cells via IGF1–IGF1R, macrophages from human peripheral blood, after incubating with PBS, trophoblast-derived extracellular vesicles from NP (NP-EVs) or PE-EVs in vitro, were co-cultured with CD4+ naive T cells (*Figure 7C*). The frequencies of memory-like CD45RO+ CCR7+ IL-17A+ CD4+ cells significantly increased in the PE-EV-induced macrophages group rather than in the NP-EV-induced macrophages group. However, the frequencies of memory-like Th17 cells significantly decreased after CD4+ naive T cells were treated with the IGF1R inhibitor BMS-754807 (*Figure 7D*).

To investigate the role of IGF1–IGF1R in the development of PE in vivo, CD45+CD4+ T cells from NP mice were isolated and treated with BMS-754807 or PBS, then injected into NP mice at day 11.5 of gestation (*Figure 7E*). Lipopolysaccharide (LPS)-induced PE mice model was constructed by intraperitoneal injection of LPS on days 12.5 and 15.5 of gestation (*Han et al., 2021*), and an anti-CD4 antibody was used to deplete the CD4+ T cells in pregnant mice. The mice injected with CD4+ T cells treated by BMS-754807 presented decreased embryo resorption rate and increased pup crown-rump length and fetal weight (*Figure 7F*), decreased SBP and UACR (*Figure 7G*), and decreased frequency of memory-like Th17 cells at the maternal–fetal interface (*Figure 7H*). These data suggested that pro-inflam Macs could induce the production of memory-like Th17 via the IGF1–IGF1R, leading to the development of PE.

## Discussion

PE, a progressive systemic disease during pregnancy, is closely related to the alterations in the immune environment at the maternal–fetal interface. In this study, the overall immune cell profiles in the placenta of individuals with NP, PE, GDM, and GDM&PE were detected by CyTOF and a PE-specific change in immune microenvironment of maternal–fetal interface was described for the first time. Further, we provided a novel insight that pro-inflam Macs induced memory-like Th17 cells, memory-like CD8+ T cells, and suppressed the production of gMDSCs in mice maternal–fetal interface by scRNA-seq. In addition, we first validated that IGF1–IGF1R was involved in producing memory-like Th17 cells induced by pro-inflam Macs in vitro and in vivo, thus leading to the development of PE.

Various studies represented the effect of a single subset of immune cells in the pathogenesis of PE or GDM (*Bachmayer et al., 2006*; *Care et al., 2018*; *Cornelius et al., 2015a*; *Eghbal-Fard et al., 2019*; *Faas et al., 2014*; *Fu et al., 2014*; *Fukui et al., 2011*; *Lampé et al., 2015*; *Lampé et al., 2011*; *Lang et al., 2021*; *Liu et al., 2021*; *Lu et al., 2020*; *Santner-Nanan et al., 2009*; *Sasaki et al., 2007*; *Travis et al., 2020*; *Yao et al., 2019*). *Miller et al., 2022* provided an overall analysis of immune cells in the human placental villi in the presence and absence of spontaneous labor at term by scRNA-seq (*Miller et al., 2022*). Luo et al. unveiled the intricate interplay existing between SPP1+M1 cells and the IGFBP1+SPP1+ extracellular villous trophoblast in the context of PE by using scRNA-seq (*Luo et al., 2023*). Li et al. highlighted a key role of a distinct perivascular inflammatory CD11chigh decidual macrophages subpopulation in the pathogenesis of PE through scRNA sequencing (*Li et al., 2024*). However, the immune cells crosstalk with each other at the maternal–fetal interface elaborately. A

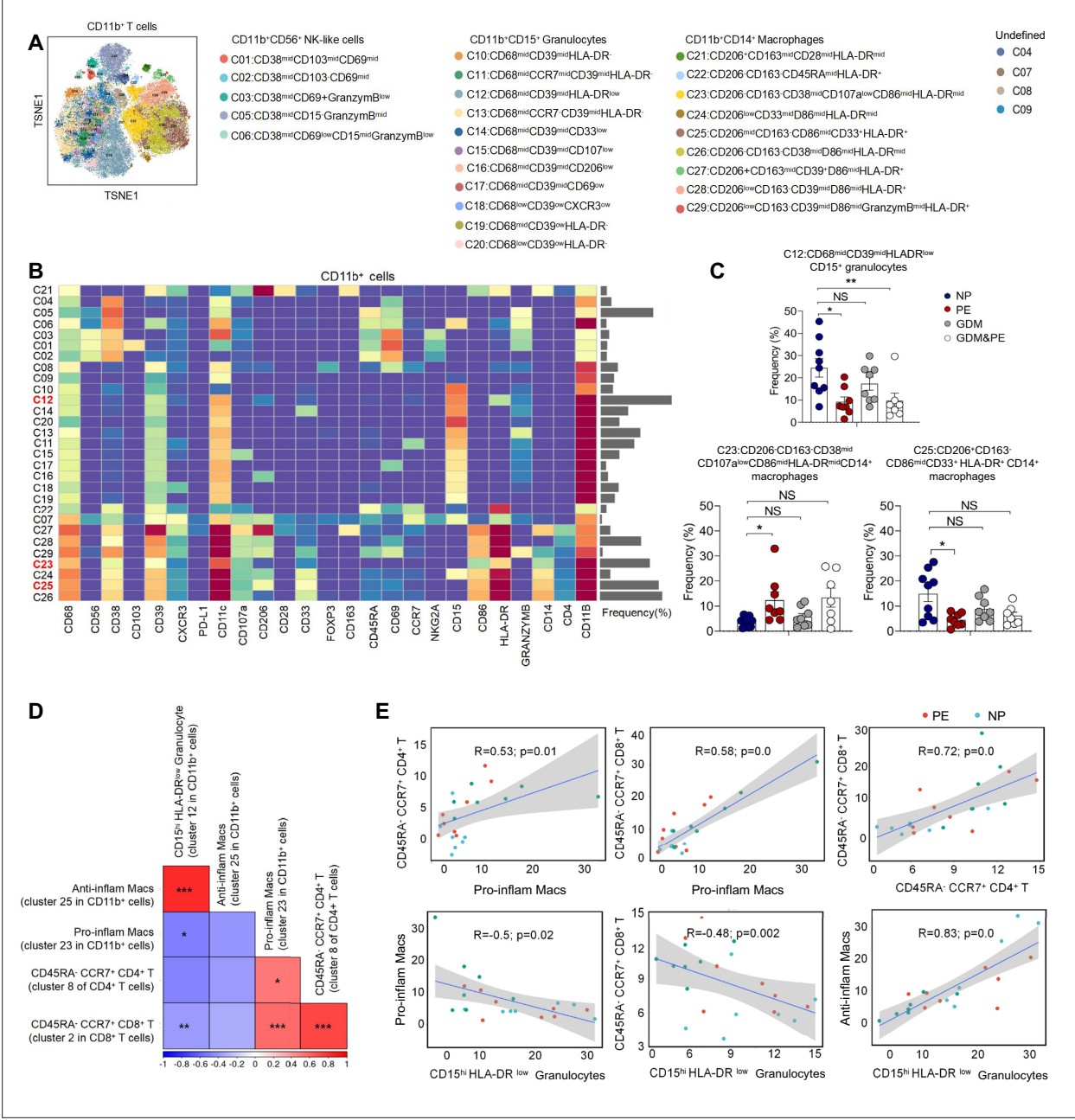

**Figure 3.** Identification of the placental CD11b+ cell subsets and the interaction between placental immune cells. (**A**) Distribution of the CD11b+ immune cells in group analyzed using t-Distributed Stochastic Neighbor Embedding (t-SNE). (**B**) Heatmap showing the expression levels of markers in the CD11b+ cells. (**C**) Scatter dot plots showing the frequencies of clusters 12, 23, and 25 of CD11b+ cells in the placentas. (**D**) Interaction between placental immune cells showed in heatmap. (**E**) Scatter plots of Pearson's correlation analysis between placental immune cells. Data were compared between NP and PE, NP and GDM, and NP and GDM&PE using the Kruskal–Wallis test and represented as mean ± SEM (*p < 0.05, **p < 0.01, ***p < 0.001; NS, not significant).

The online version of this article includes the following source data for figure 3:

**Source data 1.** Raw data and detailed analysis for **Figure 3**.

comprehensive and in-depth understanding of the immune microenvironment at the maternal–fetal interface of PE is still lacking. Here, we used CyTOF and scRNA-seq to comprehensively analyze the immune cell profile and explore the interaction of immune cells in PE. Additionally, most previous studies have reported immune cells from maternal systemic circulation rather than directly from the tissue of maternal–fetal interface (*Cornelius et al., 2015b*; *Deer et al., 2021*; *Santner-Nanan et al.,*

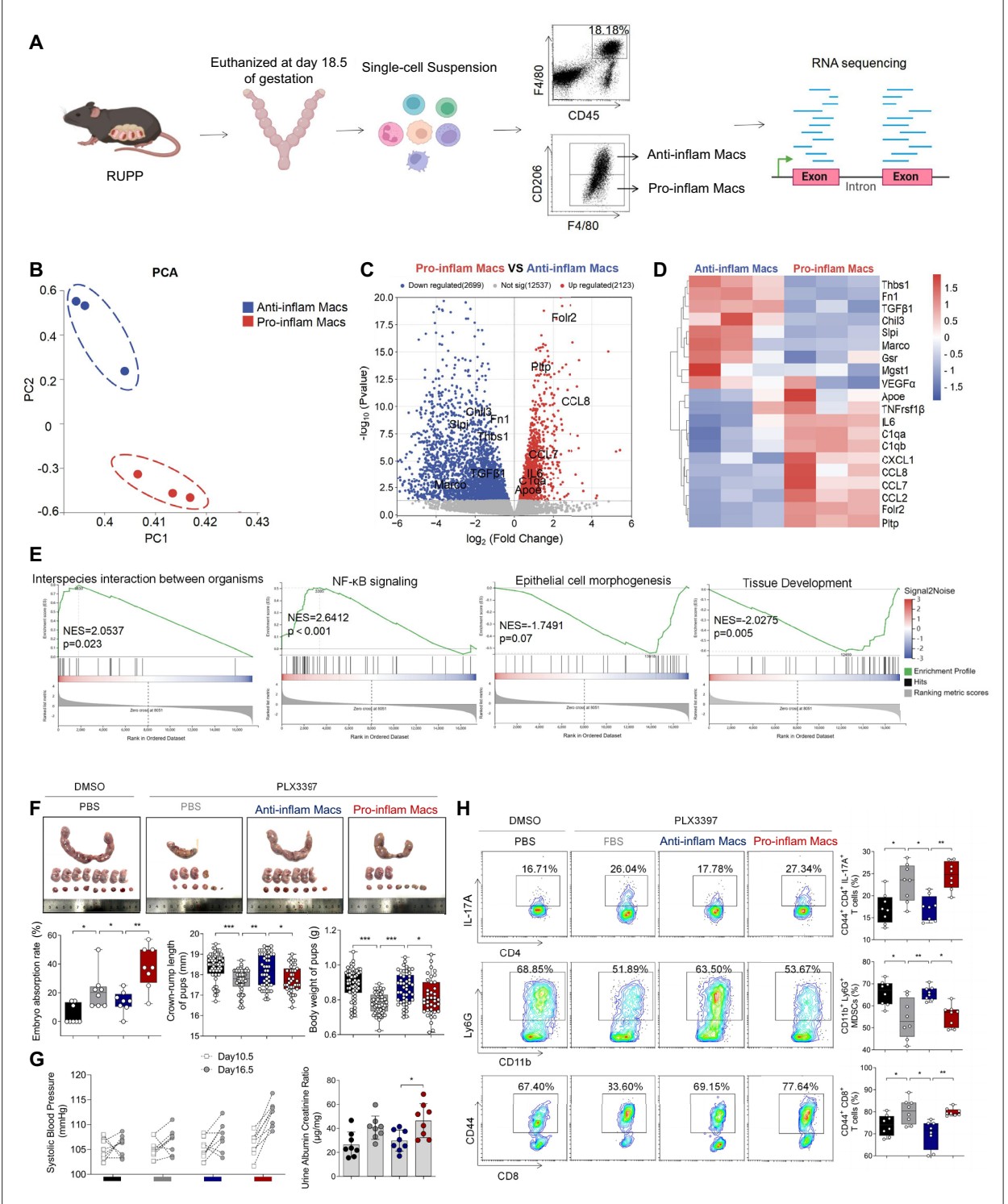

**Figure 4.** The immune imbalance at the maternal–fetal interface induced by F480⁺CD206⁻ pro-inflam Macs. (**A**) An illustration showcasing the RNA-seq of the CD45⁺F4/80⁺CD206⁻ pro-inflam Macs and CD45⁺F4/80⁺CD206⁺ anti-inflam Macs derived from the RUPP mouse model. (**B**) Principal component analysis (PCA) reflected the differences between the two groups of macrophages (*n* = 3). (**C**) The volcano map shows a comparison of the content and p value of gene expression between pro-inflam and anti-inflam Macs. Differential expression genes were screened out when p < 0.05. Red dots indicate genes with increased expression in pro-inflam Macs. Blue dots indicate genes with decreased expression. (**D**) The volcano map shows differential expression genes between pro-inflam and anti-inflam Macs. (**E**) Representative pathways enriched in the identified genes as determined by gene set enrichment analysis (GSEA) (p-value <0.05). (**F**) Embryo abortion rate of the pregnant mice, body weight and crown-rump length of pups measured

*Figure 4 continued on next page*

*Figure 4 continued*

on day 18.5 of gestation. Black represents mice treated with DMSO ($n = 8$); gray represents mice treated with PLX3397 ($n = 8$); blue represents mice injected with CD45$^+$F4/80$^+$CD206$^+$ anti-inflammatory macrophages ($n = 8$); red represents mice injected with CD45$^+$F4/80$^+$CD206$^-$ pro-inflammatory macrophages ($n = 8$). (G) Systolic blood pressure (SBP) and UACR of pregnant mice in the four groups. (H) Frequencies of CD44$^+$CD4$^+$IL-17A$^+$ cells, CD44$^+$CD8$^+$ T cells, and CD11b$^+$Ly6G$^+$ granulocytes analyzed by flow cytometry. Data were compared between groups using the Kruskal–Wallis test and represented as mean ± SEM (*$p < 0.05$, **$p < 0.01$, ***$p < 0.001$). RNA transcriptome soure data are stored in Dryad Digital Repository, doi:10.5061/dryad.4qrfj6qn0.

The online version of this article includes the following source data and figure supplement(s) for figure 4:

**Source data 1.** Raw data and detailed analysis for *Figure 4*.

**Figure supplement 1.** The construction of RUPP mouse model.

**Figure supplement 1—source data 1.** Raw data and detailed analysis for *Figure 4—figure supplement 1*.

**Figure supplement 2.** Clodronate liposomes were used to deplete the macrophages of pregnant mice to demonstrate that pro-inflam Macs lead to immune imbalance.

**Figure supplement 2—source data 1.** Raw data and detailed analysis for *Figure 4—figure supplement 2*.

*2009*; *Shields et al., 2018*; *Wallace et al., 2012*). In this study, cells were directly isolated from the tissue of maternal–fetal interface, which is considered as a more accurate approach to unveil the real immune environment of the maternal–fetal interface.

Abnormal inflammatory responses were reported to play an important role in the pathogenesis of both PE and GDM (*Aneman et al., 2020*; *Corrêa-Silva et al., 2018*; *Deer et al., 2023*; *Jung et al., 2022*; *McElwain et al., 2021*). However, it is unclear how the maternal–fetal interface immune microenvironment differs between the two diseases. We analyzed the overall placental immune cell profile in NP, PE, GDM, and GDM&PE by CyTOF and revealed a PE-specific immune cell profile: The frequencies of memory-like Th17 cells (CD45RA$^-$CCR7$^+$IL-17A$^+$CD4$^+$), memory-like CD8$^+$ T cells (CD45RA$^-$CCR7$^+$CD38$^+$pAKT$^{mid}$CD127$^{low}$CD8$^+$) and pro-inflam Macs (CD206$^-$CD163$^-$CD38$^{mid}$CD107a$^{low}$CD86$^{mid}$HLA-DR$^{mid}$CD14$^+$) were increased, while the decreased frequencies of CD69$^{hi}$Helios$^{mid}$CD127$^{mid}$ γδT cells, anti-inflam Macs (CD206$^+$CD163$^-$CD86$^{mid}$CD33$^+$HLA-DR$^+$CD14$^+$) and gMDSCs (CD11b$^+$CD15$^{hi}$HLA-DR$^{low}$) were observed in the placenta of PE, but not in that of GDM or GDM&PE compared with that of NP.

PE is often divided into two subtypes based on the time of onset. Late-onset PE, occurring at 34 weeks of gestation or later, accounts for most PE cases, while early-onset PE, occurring before or at 33 weeks of gestation (*Lisonkova and Joseph, 2013*). However, our previous study observed no significant difference in the M1/M2 macrophage polarization in placentas with early- or late-onset PE (*Liu et al., 2022*). Therefore, our research was conducted regardless of the PE-onset type to ensure consistency in macrophage polarization of the two subtypes.

There have been studies indicating that pro-inflammatory macrophages, Th17 and CD8$^+$ T cells play an essential role in PE development (*Care et al., 2018*; *Eghbal-Fard et al., 2019*; *Lager et al., 2020*; *Yao et al., 2019*). Macrophages was indicated with a pro-inflammatory phenotype in the placenta with PE (*Faas et al., 2014*; *Yao et al., 2019*). *Care et al., 2018* reported that placental Tregs helped suppress inflammation at the maternal–fetal interface. Similarly, *Lu et al., 2020* found an abnormal increase of IL-17A in the placenta in the development of PE. Meanwhile, the abnormal infiltration of CD8$^+$ T cells in the placenta may contribute to PE (*Lager et al., 2020*). Consistent with the previous findings, we found that memory-like CD4$^+$ and memory-like CD8$^+$ T cells significantly increased in the PE group and that IL-17A was significantly higher expressed in CD4$^+$ memory-like T cells from the PE group, which is vital in vascular inflammation, leading to the development of PE (*Amador et al., 2014*; *Madhur et al., 2021*). Though memory T cells play a key role in fetal–maternal tolerance in NP (*Kieffer et al., 2019*), few studies have reported the characteristics of memory-like T cells in the development of PE.

We constructed an adoptive transferred mouse model. The injection of CD4$^+$ T cells from RUPP mouse, characterized by an increased frequency of Th17 cells and a reduced frequency of Tregs, induced PE-like symptoms in pregnant mice. Tregs which are well known to undergird maternal tolerance of the fetus, and which are well known to have an overlapping developmental trajectory with RORgt$^+$ Th17 cells (*Robertson et al., 2019*; *Sefik et al., 2015*). This study underscores the importance of Th17/Treg cell balance in maintaining pregnancy homeostasis. And it is worth noting that

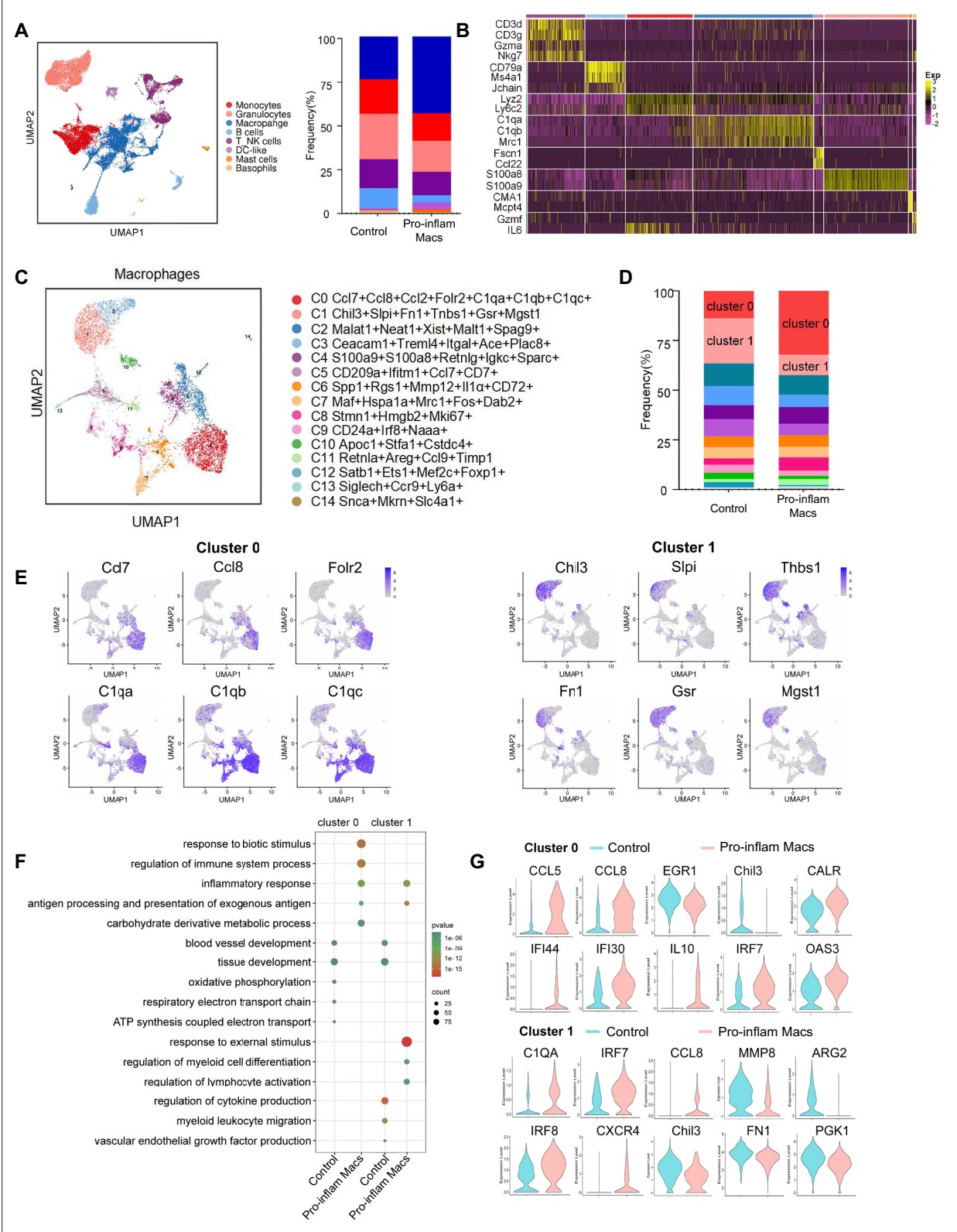

**Figure 5.** Identification of the macrophage subsets in mice injected with pro-inflam and anti-inflam Macs by using single-cell RNA sequencing (scRNA-seq). (**A**) Uniform Manifold Approximation and Projection (UMAP) maps showing the eight types of mouse immune cells at the maternal–fetal interface. (**B**) Heatmap showing clustering analysis for markers distinguished different type of immune cells. (**C**) UMAP maps showing the 15 clusters of mouse macrophages was listed. (**D**) Bar graph showing the frequencies of clusters of macrophages in the two groups of mice was listed in the right panel. (**E**)

*Figure 5 continued on next page*

*Figure 5 continued*

UMAP maps showing the distribution of specific markers of clusters 0 and 1. (**F**) Dot plot depicting GO enrichment terms that were significantly enriched in the differentially expressed genes in clusters 0 and 1 from the pro-inflam Macs group and the control group. (**G**) Violin plot of specific differential gene expression in clusters 0 and 1 between the pro-inflam Macs and the control groups. Single-cell RNA transcriptome soure data are stored in Dryad Digital Repository, 10.5061/dryad.4qrfj6qn0.

The online version of this article includes the following source data and figure supplement(s) for figure 5:

**Source data 1.** Raw data and detailed analysis for *Figure 5*.

**Figure supplement 1.** Heatmap of different clusters of macrophages.

**Figure supplement 1—source data 1.** Raw data and detailed analysis for *Figure 5—figure supplement 1*.

---

we isolated CD4[+] memory-like T cells directly from the maternal–fetal interface, which is different from previous studies that the CD4[+] T cells for adoptive transfer were isolated from the spleen and induced in vitro (*Cornelius et al., 2015b*; *Deer et al., 2021*; *Shields et al., 2018*; *Wallace et al., 2012*). Recurrence is an important concern in individuals with PE (*Bernardes et al., 2019*). We found that a prior PE pregnancy increased the probability of PE in the next pregnancy in mice, accompanied by an increased frequency of memory-like Th17 cells at the maternal–fetal interface, suggesting that increased memory-like Th17 cells are positively related to the recurrence of PE. However, further studies are needed to confirm the function of the fetal-specific memory Th17 cells in the recurrence of PE.

Activation of CD8[+] T and Th1 cells can be induced by abnormal polarized macrophages in recurrent miscarriage (*Li et al., 2022a*). The gMDSCs have been reported to have immune suppressive activity and are necessary for maintaining maternal–fetal tolerance (*Köstlin-Gille et al., 2019*; *Zhou et al., 2018*). However, the correlation between macrophages and other immune cells in the development of PE remained unclear. We verified that pro-inflam Macs induced memory-like Th17 cells and memory-like CD8[+] T cells and inhibited the production of gMDSCs in PE by correlation analysis and adoptive transfer in animal experiments for the first time, indicating that the immune cells orchestrate a network in the maternal–fetal interface elaborately. However, PLX3397, used in this study, emerges as a promising small-molecule compound, endowed with the capability to traverse the placental barrier, thereby potentially influencing the development and function of fetal macrophages. In addition to gMDSCs, we conducted immunofluorescence analysis of placental tissue, specifically targeting CD66b[high] neutrophils. Our findings revealed a pronounced increase in the proportion of neutrophils among PE patients, fostering the hypothesis that IL-17A produced by Th17 cells might orchestrate the migration of neutrophils toward the placental milieu. This migratory response, in turn, could potentially trigger reactive oxygen species-mediated immunopathology, which subsequently lead to PE (*Akkari et al., 2024*).

The function of IGR1 and IGF1R in PE is controversial. IGF1 plays a significant role in maintaining immune function, and IGF1R facilitates the differentiation of naive CD4[+] T into Th17 cells (*DiToro et al., 2020*). A study showed that IGF1R was downregulated in the serum of individuals with PE (*Liao et al., 2021*). Decreased amounts of IGF1R were also found in the placentas of individuals with PE (*Robajac et al., 2015*). However, another study found no significance in the affinity and the number of IGF1R between placentas from individuals with PE or not (*Díaz et al., 2005*). We found that mice exhibiting PE symptoms had significantly enhanced IGF1–IGF1R interaction between inflammatory macrophages and memory-like Th17 cells by analyzing the data of scRNA-seq. Then, this finding was further confirmed in human placentas from individuals with PE and NP, for the frequencies of IGF1[+] CD14[+] and IGF1R[+] CD4[+] cells were found to significantly increase in the PE group. In vitro, we found that PE-EV-treated macrophages secreted more IGF1 than NP-EV-treated macrophages and induced a higher frequency of memory-like Th17 cells. The inhibition of IGF1R on CD4[+] naive T cells resulted in a decreased frequency of memory-like Th17 cells, implying that the IGF1–IGF1R is critical for the production of memory-like Th17 cells induced by inflammatory macrophages. Animal experiments confirmed that IGF1R inhibition on CD4[+] T cells at the maternal–fetal interface reduced the frequency of memory-like Th17 cells and relieved PE-like symptoms.

In addition, placentas from different groups for CyTOF analysis and flow cytometry were matched by age, preterm pregnancies, and previous abortion history; however, there are differences in body mass index, gestational age, term pregnancies, and the number of living children. Obesity can increase

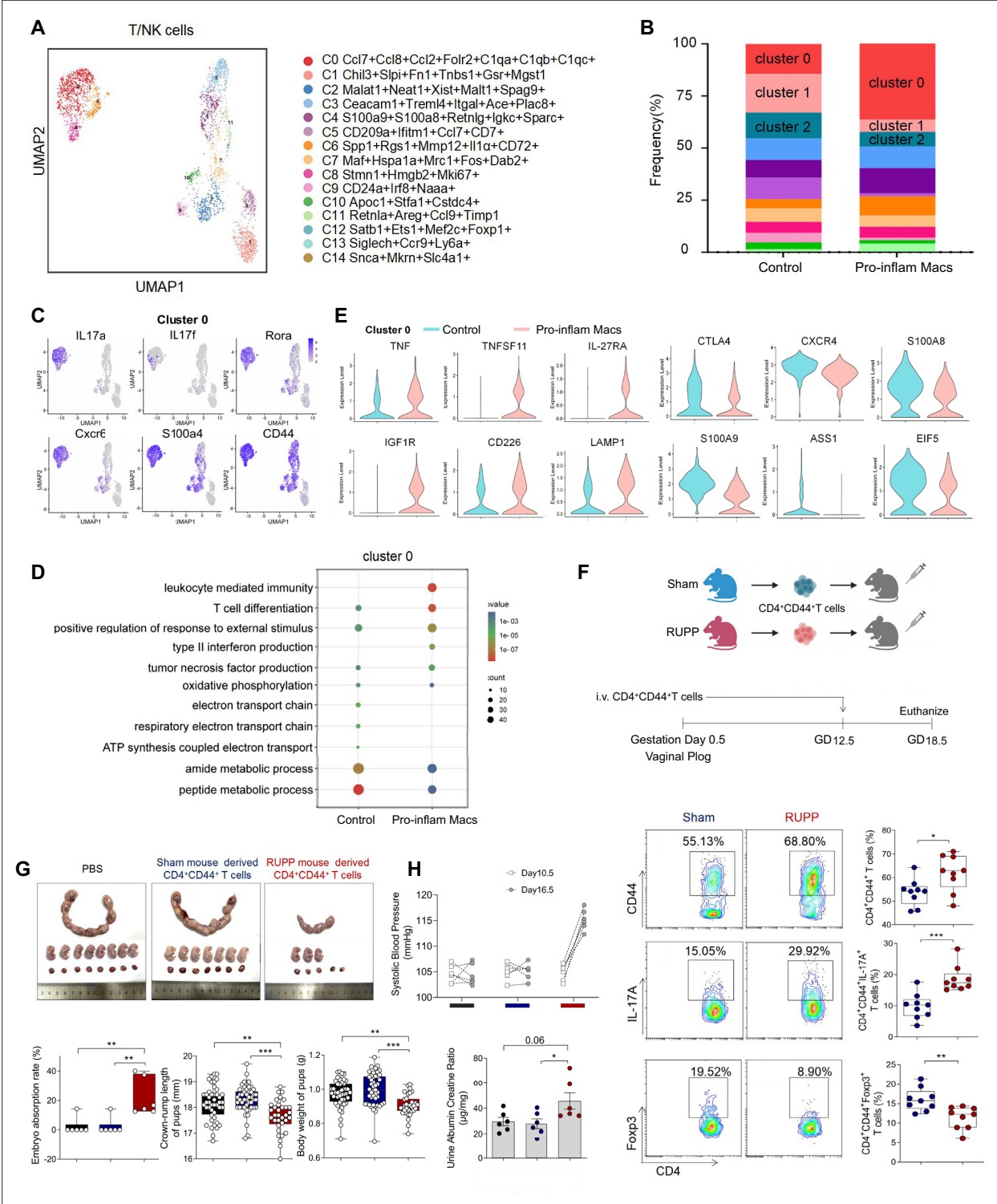

**Figure 6.** Identification of the T/NK cells subsets in mice injected pro-inflam and anti-inflam Macs using single-cell RNA sequencing (scRNA-seq). (**A**) Uniform Manifold Approximation and Projection (UMAP) maps showing the 12 clusters of mouse T/NK cells was listed. Heatmap showing clustering analysis for markers distinguished 12 different clusters of T/NK cells. (**B**) Bar graph showing the frequencies of clusters of T/NK cells in the two groups of mice was listed in the down panel. (**C**) Dot plot depicting GO enrichment terms that were significantly enriched in the differentially expressed genes in cluster 0 from the pro-inflam Macs and the control groups. (**D**) Violin plot of specific differential gene expression in cluster 0 between the pro-inflam Macs and the control groups. (**F**) Frequencies of CD4+CD44+ T cells and the percentages of Foxp3+ or IL-17A+ cells in CD4+CD44+ T cells at the maternal–fetal interface in Sham and RUPP group analyzed by flow cytometry. (**G**) The embryo abortion rate of pregnant mice, body weight, and

*Figure 6 continued on next page*

*Figure 6 continued*

crown-rump length of pups measured on day 18.5 of gestation in mice injected PBS, Sham mouse-derived or RUPP mouse-derived CD4$^+$CD44$^+$ T cells. Black represents mice injected with PBS (*n* = 6); blue represents mice injected with Sham mouse-derived CD4$^+$CD44$^+$ T cells (*n* = 6); red represents mice injected with RUPP mouse-derived CD4$^+$CD44$^+$ T cells (*n* = 6). (**H**) Systolic blood pressure (SBP) and UACR of pregnant mice injected with PBS, Sham mouse-derived or RUPP mouse-derived CD4$^+$CD44$^+$ T cells. Data were compared between groups using one-way ANOVA and represented as mean ± SEM (*p < 0.05, **p < 0.01, ***p < 0.001). Data were compared between the two groups using the Student's *t*-test and represented as mean ± SEM (*p < 0.05, **p < 0.01, ***p < 0.001).

The online version of this article includes the following source data and figure supplement(s) for figure 6:

**Source data 1.** Raw data and detailed analysis for *Figure 6*.

**Figure supplement 1.** Memory-like Th17 cells may be associated with the recurrence of PE.

**Figure supplement 1—source data 1.** Raw data and detailed analysis for *Figure 6—figure supplement 1*.

**Figure supplement 2.** Identification of the granulocytes subsets in mice injected pro-inflam and anti-inflam Macs use single-cell RNA sequencing (scRNA-seq).

**Figure supplement 2—source data 1.** Raw data and detailed analysis for *Figure 6—figure supplement 2*.

inflammatory and oxidative stress markers in the placental environment (*Spradley et al., 2015*). Also, it has been reported that the placental immune state shifts with gestational age (*Lewis et al., 2018*). However, as PE is often accompanied by obesity and early termination of pregnancy, it is difficult to exclude these factors in sample collection. Moreover, limited placental samples in the GDM&PE group are the shortage of this study, for it is hard to collect enough clean samples that exclude interference factors because the number of pregnant women exposed to COVID-19 has increased sharply since December 2022 in China.

In summary, this study demonstrated the statistically distinguished placental immune microenvironment in individuals with PE, but not in those with GDM or GDM&PE. More importantly, macrophages orchestrate a network in the maternal–fetal interface elaborately. These findings provide novel insights into PE pathogenesis and a potential immune target for the clinical prevention and treatment of PE.

## Materials and methods
### Clinical sample collection

The samples used in this study were collected from Sir Run Run Shaw Hospital between October 2020 and August 2023. Informed consent was obtained from all volunteers, and the Ethics Committee of Sir Run Run Shaw Hospital, Zhejiang University School of Medicine, approved the study. Human placentas were obtained from women with NP, PE, GDM, or GDM&PE who underwent elective cesarean delivery. The diagnostic criteria for PE included new-onset hypertension after 20 weeks of gestation with SBP ≥140 mmHg and/or diastolic blood pressure ≥90 mmHg and proteinuria (≥300 mg) on at least two occasions. A positive glucose tolerance test diagnoses GDM. Women with normal blood pressure, full-term pregnancies, and no complications were designated as controls. The detailed clinical characteristics of the pregnant women in this study are presented in *Tables 5 and 6*.

### Laboratory mice

Eight-week-old female C57 mice and ten-week-old male BALB/c mice were purchased from Hangzhou Ziyuan Laboratory Animal Technology Co, Ltd (Hangzhou, China) and Shanghai Jihui Experimental Animal Breeding Co, Ltd (Shanghai, China), respectively. All animals were maintained under pathogen-free conditions. The Guide for the Care and Use of Laboratory Animals (China) conducted all experimental procedures involving animals, and the Animal Research Ethics Committee of the Sir Run Run Shaw Hospital of Zhejiang University approved the protocols.

Female C57 mice were mated with male BALB/c mice to establish an allogeneic pregnancy model (*Rowe et al., 2012*). The day of the vaginal plug detection was considered day 0.5 of pregnancy. SBP was measured using a noninvasive mouse tailcuff BP analyzer (BP-2010A, Softron, Japan) at 12.5, and 16.5 days of gestation. Three random urine samples were collected after day 16.5 of gestation, and UACR was measured by urinary microalbumin (CH0101060, Maccura, China). Mice were euthanized on day 18.5 of gestation.

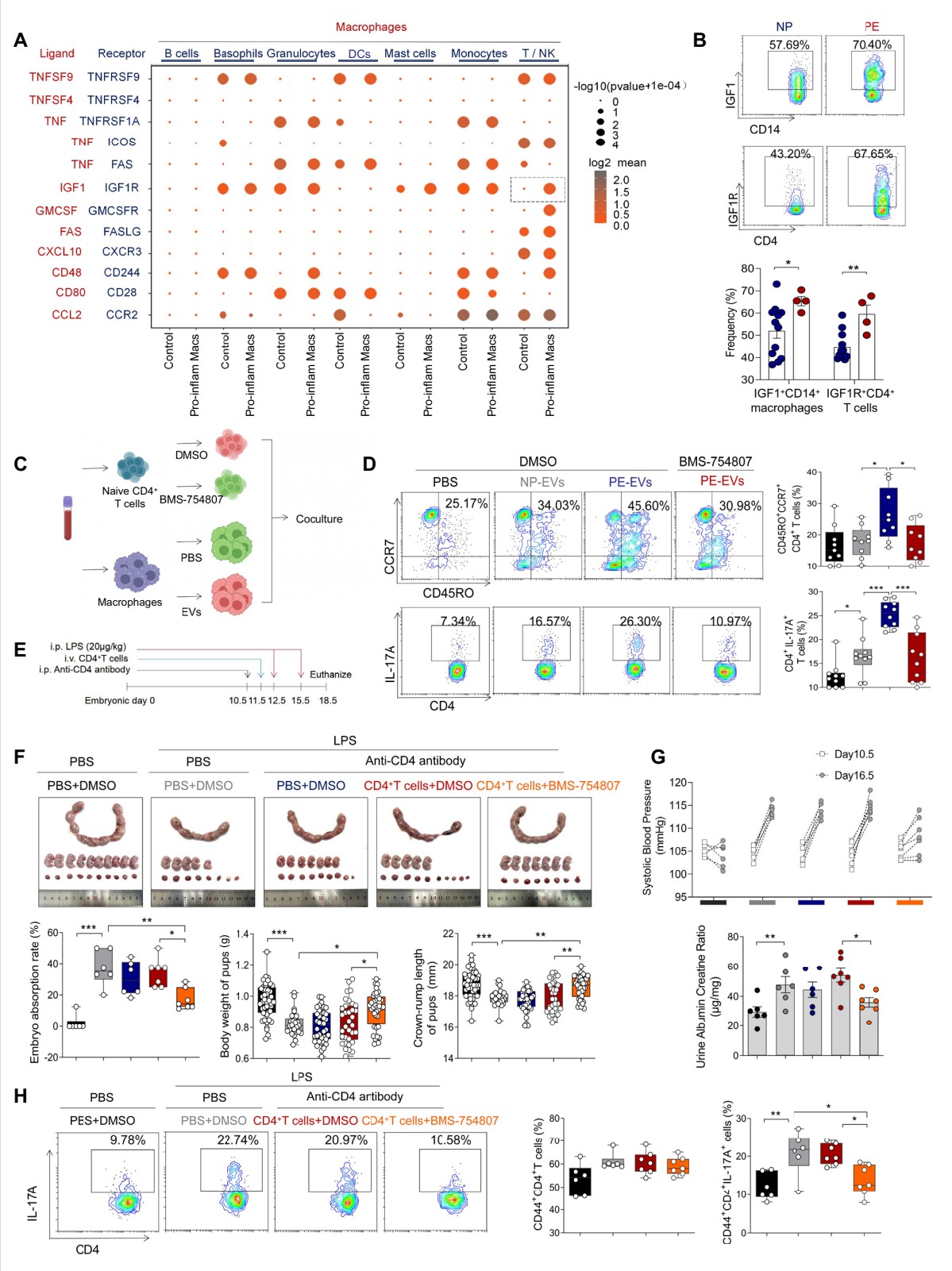

**Figure 7.** Pro-inflam Macs induce the generation of memory-like Th17 cells via IGF1–IGF1R. (**A**) Signaling modules indicated by ligand–receptor pairing between macrophages and other types of immune cells at the maternal–fetal interface using CellPhoneDB. (**B**) Frequencies of IGF1⁺CD14⁺ and IGFIR⁺CD4⁺ cells in placentas of individuals with NP and PE (*n* = 12 in NP group, *n* = 4 in PE group). (**C**) Schematic of the experimental workflow to induce memory-like T cells in vitro. Macrophages, after incubating with PBS, NP-EVs or PE-EVs, were co-cultured with CD4⁺ naive T cells treated

*Figure 7 continued on next page*

*Figure 7 continued*

with DMSO or BMS-754807. Cells were isolated from human peripheral blood. (**D**) Frequencies of CD45RO⁺CCR7⁺Th17 cells. Black represents CD4⁺ naive T cells treated with DMSO; gray represents CD4⁺ naive T cells co-cultured with NP-EV-treated macrophages; blue represents CD4⁺ naive T cells co-cultured with PE-EV-treated macrophages; red represents CD4⁺ naive T cells treated with BMS-754807 before co-cultured with PE-EV-treated macrophages (*n* = 10 in each group). (**E**) Schematic of mice transferred CD4⁺ T cells treated with BMS-754807 or PBS. Anti-CD4 antibody was used to deplete CD4⁺ T cells in mice on day 10.5 of gestation. CD4⁺ T cells were transferred into mice on day 11.5 of gestation. 20 µg/kg lipopolysaccharide (LPS) was intraperitoneally injected on days 12.5 and 15.5 of gestation to induce a PE-like pregnant mice model. Mice were sacrificed on day 18.5 of gestation. (**F**) Embryo abortion rate of pregnant mice, body weight and crown-rump length of pups were measured on day 18.5 of gestation. Black represents the control group mice (*n* = 6); gray represents mice treated with LPS (20 µg/kg) to construct an animal model of PE (*n* = 6); blue represents anti-CD4 antibody treated PE mice (*n* = 6); red represents anti-CD4 antibody treated PE mice injected with CD4⁺ T cells with DMSO treatment (*n* = 7); orange represents anti-CD4 antibody treated PE mice injected with CD4⁺ T cells with BMS754807 treatment (*n* = 7). (**G**) Systolic blood pressure (SBP) and UACR of pregnant mice in the five groups. (**H**) The frequencies of CD4⁺ CD44⁺ IL-17A⁺ cells analyzed by flow cytometry. Data were compared between groups using one-way ANOVA and represented as mean ± SEM (*$p < 0.05$, **$p < 0.01$, ***$p < 0.001$).

The online version of this article includes the following source data and figure supplement(s) for figure 7:

**Source data 1.** Raw data and detailed analysis for *Figure 7*.

**Figure supplement 1.** Cell–cell communications in immune cells when pro-inflam Macs accumulated at the maternal–fetal interface.

**Figure supplement 1—source data 1.** Raw data and detailed analysis for *Figure 7—figure supplement 1*.

## Reduction in uterine perfusion pressure mouse model

To construct a mouse model of PE, we ligated uterine arteries in pregnant mice on day 12.5 of gestation. Briefly, after 4% chloral hydrate was intraperitoneally injected for anesthesia, bilateral incisions were made on the back of the pregnant mice, and surgical sutures were used to reduce the blood flow of the bilateral uterine arcades.

## Isolation of single cells from the mouse uterus and human placenta

Uterus from pregnant mice and placentas from volunteers were washed twice with ice-cold PBS and cut into small pieces. The tissues were digested with collagenase type IV (1 mg/ml, Sigma-Aldrich, USA) and DNase I (0.01 mg/ml, Sigma-Aldrich, U.S.A) in RPMI 1640 medium (Thermo Fisher Scientific) for 40 min at 200 rpm and 37°C. The suspensions were strained through 70-µm nylon mesh and centrifuged at 500 × *g* for 5 min. Leaving the supernatants, the cell pellets from human placentas need an extra purification by Ficoll (P4350, Solarbio, China) according to the manufacturer's instructions. Human CD4⁺ memory T cells were isolated using a Human Central and Effector Memory CD4⁺ T Cell Isolation Kit (17865, STEMCELL, Canada).

## Adoptive transferred mouse model

Endogenous macrophages or CD4⁺ T cells were depleted by injecting PLX3397 (S7818, Selleck, USA), clodronate liposomes (40337ES08, YEASEN, China), or anti-CD4 antibody (BE0003-1, BioXcell, USA) intraperitoneally every 3 days starting from day 10.5 of gestation.

For the adoptive transferred mouse model, phycoerythrin-conjugated anti-mouse CD4 (12-0041-82, eBioscience, USA) and Pc5.5-conjugated anti-mouse CD44 (45-0441-82, eBioscience, USA) were used to label CD4⁺ memory-like T cells from uterus and placentas. PE-conjugated anti-mouse CD45 (E-AB-F1136D, Elabscience, China), FITC-conjugated anti-mouse F4/80 (11-4801-82, eBioscience, USA), and PE-Cyanine7-conjugated anti-mouse CD206 (141719, BioLegend, USA) were used to label anti-inflam Macs from uterus and placentas, and the remaining CD45⁺F480⁺CD206⁻ cells were considered pro-inflam Macs. The MoFlo Astrios EQ FACS Cell Sorter at Core Facilities, ZheJiang University

**Table 5.** Details of the individual with NP or PE included in the study.

| Parameters | NP (*n* = 30) | PE (*n* = 30) | p value |
|---|---|---|---|
| Age (years) | 31.13 ± 2.9747 | 31.50 ± 4.1693 | 0.0741 |
| BMI (kg/m²) | 27.24 ± 2.8048 | 29.48 ± 3.9295 | 0.0155 |
| Gestational age (weeks) | 38.68 ± 0.6038 | 35.53 ± 3.29 | <0.0001 |
| Number of living children | 0.6000 ± 0.4900 | 0.4667 ± 0.6182 | 0.3665 |

**Table 6.** Details of the individual with early- or late-onset preeclampsia included in the study.

| Parameters | Early-onset preeclampsia (*n* = 11) | Late-onset preeclampsia (*n* = 19) | p value |
|---|---|---|---|
| The gestational week when blood pressure begins to rise | 28.82 ± 2.8220 | 36.05 ± 1.129 | <0.0001 |
| Age (years) | 32.45 ± 4.6768 | 32.36 ± 3.8472 | 0.9568 |
| BMI (kg/m²) | 29.64 ± 2.6840 | 28.71 ± 4.4506 | 0.5388 |
| Gestational age (weeks) | 33.45 ± 3.7887 | 36.77 ± 1.2171 | 0.0014 |
| Number of living children | 0.36 ± 0.5045 | 0.36 ± 0.4956 | 0.98 |

School of Medicine (China), was used to obtain specific cells with a purity of >96%. Briefly, $2–5 \times 10^5$ CD4$^+$ CD44$^+$ T cells, $5 \times 10^5$ pro-inflam Macs, or $5 \times 10^5$ anti-inflam Macs were injected to pregnant C57 mice intravenously at day 12.5 of gestation.

$2–5 \times 10^5$ CD45$^+$CD4$^+$ T cells from uterus and placentas of pregnant mice were sorted and transferred to pregnant C57 mice at day 11.5 of gestation after treating with 100nM BMS-754809 (HY10200; MedChemExpress; USA) or PBS for three days in vitro.

## Mass cytometry by time of flight

Placental villi rather than chorionic plate and extraplacental membranes were used for CyTOF in this study. $8 \times 10^5$ immune cells from NP (*n* = 9), PE (*n* = 8), GDM (*n* = 8), and GDM&PE (*n* = 7) presented in *Table 1* were washed once with 1× PBS and then stained with 100 µl of 2.5 µM cisplatin (Fluidigm) for 5 min on ice to exclude dead cells, and then incubated in Fc receptor blocking solution before stained with surface antibodies cocktail for 30 min on ice. Cells were washed twice with FACS buffer (1× PBS + 0.5% BSA) and fixed in 200 µl of intercalation solution (Maxpar Fix and Perm Buffer containing 250 nM 191/193Ir, Fluidigm) overnight. After fixation, cells were washed once with FACS buffer and then perm buffer (eBioscience), stained with intracellular antibodies cocktail for 30 min on ice. Cells were washed and resuspend with deionized water, adding into 20% EQ beads (Fluidigm), acquired on a mass cytometer (Helios, Fluidigm). Additionally, all PE samples enumerated in *Table 1* demonstrate a late-onset PE, with placental specimens being procured from patients more than 35 weeks of gestation and less than the 38 weeks of pregnancy.

As for data analysis, data of each sample were debarcoded from raw data using a doublet-filtering scheme with unique mass-tagged barcodes firstly. Each .fcs file generated from different batches were normalized through bead normalization method. Then we manually gate data using a FlowJo software to exclude to debris, dead cells and doublets, leaving live, single immune cells. Subsequently we applied the Phenograph clustering algorithm to all cells to partition the cells into distinct phenotypes based on marker expression levels. Cell type of each cluster according to its marker expression pattern on a heatmap of cluster vs marker was annotated. Finally, we used the dimensionality reduction algorithm t-SNE to visualize the high-dimensional data in two dimensions and show distribution of each cluster and marker expression and difference among each group or different sample type.

## RNA sequencing

$5 \times 10^5$ CD45$^+$F4/80$^+$CD206$^-$ pro-inflam Macs (*n* = 3) and $5 \times 10^5$ CD45$^+$F4/80$^+$CD206$^+$ anti-inflam Macs (*n* = 3) were isolated from the uterus and placentas of mice with RUPP and high throughput sequencing and bioinformatics analyses were conducted at Shenzhen Huada Gene Technology Service Co Ltd (Shenzhen, China). Only differential genes with more than twofold change and a corrected p value less than 0.05 were considered statistically significant.

## Single-cell RNA sequencing

Mice were injected with $5 \times 10^5$ CD45$^+$F4/80$^+$CD206$^-$ pro-inflam Macs or $5 \times 10^5$ CD45$^+$F4/80$^+$CD206$^+$ anti-inflam Macs at 12.5 days of gestation and euthanized on day 18.5 of gestation. CD45$^+$ immune cells from the uterus and placenta were isolated from mice transferred pro-inflam and anti-inflam Macs. scRNA-seq was conducted at PLTTECH Service Co Ltd (Hangzhou, China).

First, the outcomes of cellular high-throughput sequencing were preserved in the FASTQ file format, facilitating a rigorous evaluation of the raw data. Subsequently, the data underwent debarcoding,

enabling its alignment with the reference genome to precisely identify and sieve effective cells. This process culminated in the construction of a comprehensive gene-barcode expression matrix. For the statistical analysis and visualization of gene expression levels and Unique Molecular Identifier counts, we leveraged the powerful Seurat software. Furthermore, we conducted a clustering analysis rooted in the gene-barcode expression matrix, utilizing Seurat's capabilities in conjunction with principal component analysis for dimensionality reduction. This allowed us to scale down the clustering results and visually represent them in two dimensions using either Uniform Manifold Approximation and Projection or t-SNE. To gain deeper insights, we performed Marker genes analysis, which hinged on the clustering outcomes. This enabled us to pinpoint the genes that significantly distinguished each subgroup, thereby identifying their characteristic genes. Additionally, we analyzed and graphically represented the varying expression levels of these genes across different groups, providing a comprehensive understanding of their differences.

## Isolation of NP-EVs and PE-EVs

NP-EVs and PE-EVs were isolated and identified using our published protocols (*Jiang et al., 2021*; *Liu et al., 2022*). Briefly, after digesting the placental tissues from NP or PE, the suspensions were filtered through 100-μm nylon mesh and centrifugated at $3000 \times g$ for 15 min. Then the supernatants were filtered with a 0.22-μm filter and centrifuged at $100,000 \times g$ for 1 hr at 4°C. Then the pellets of EVs were resuspended and centrifuged at $100,000 \times g$ once again. A BCA assay kit conducted Protein quantitation of EVs (23235, Thermo Fisher Scientific). To isolate the T-EVs, 500 μg of the EVs were incubated with 1 μg placental alkaline phosphatase antibody (SC-47691, Santa Cruz Biotechnology) at 4°C overnight, then washed in the recommended buffer (PBS containing 2% exosome-free FBS and 1 mmol/l ethylenediaminetetraacetic acid), and centrifuged at $100,000 \times g$ for 1 hr at 4°C. After resuspended, the total EVs were sorted by EasySep Mouse PE Positive Selection Kit II (17666, STEMCELL) to collect the final NP-EVs or PE-EVs.

## Induction of CD4$^+$ memory-like T cells

Macrophages were obtained after 5 days of culture of mononuclear cells isolated from human peripheral blood following previous protocols (*Liu et al., 2022*). NP-EVs or PE-EVs in a 50 μg/ml concentration were added to macrophages and co-cultured at 37°C for 8 hr. The human CD4$^+$ naive T cells Isolation Kit (19555, STEMCELL, Canada) was used to isolate purified CD4$^+$ naive T cells from human peripheral blood cells according to the manufacturer's instructions. Then, CD4$^+$ naive T cells were co-cultured with EV-treated macrophages for 6 days. Flow cytometry was performed to measure the frequency of memory-like Th17 cells. For animal experiments, CD4$^+$ naive T cells sorted from the uterus and placentas of pregnant mice were cultured with the IGF1R inhibitor BMS-754807 at a concentration of 10 μM for 3 days before co-cultured with macrophages.

## Flow cytometry

Single immune cells from the mice uterus were obtained following the method described above. After incubation with Cell Stimulation Cocktail (00-4975-93, Invitrogen, USA) for 5 hr at 37°C, cells were collected and surface staining for either phycoerythrin-conjugated anti-mouse CD4 (12-0041-82, eBioscience, USA), APC-conjugated anti-mouse CD8 (100711, BioLegend, USA), phycoerythrin-Cy5.5-conjugated anti-mouse CD44 (45-0441-82, eBioscience, USA), APC-conjugated anti-mouse CD11b (101212, BioLegend, USA), FITC-conjugated anti-mouse Gr-1 (108406, BioLegend, USA), PE-Cyanine7-conjugated anti-mouse Ly6G (E-AB-F1108H, elabscience, China), or intracellular staining for 488-conjugated anti-mouse IL-17A (506910, BioLegend, USA) was performed according to the manufacturer's instructions.

Single placental and peripheral lymphocytes and were obtained following the method described in previous protocols (*Liu et al., 2022*). The Human Central and Effector Memory CD4$^+$ T cell Isolation Kit (17865, STEMCELL, Canada) was used to obtain purified memory CD4$^+$ T cells, according to the manufacturer's instructions. After incubation with Cell Stimulation Cocktail (00-4975-93, Invitrogen, USA) for 5 hr at 37°C, cells were collected and surface staining for phycoerythrin-conjugated anti-human CD4 (2384240, eBioscience, USA), FITC-conjugated anti-human CD45RO (304204, BioLegend, USA), and phycoerythrin-Cy7-conjugated anti-human CCR7 (353227, BioLegend, USA) or intracellular

staining for APC-conjugated anti-human IL-17A (17-7179-42, eBioscience, USA) and Foxp3 (320014, BioLegend, USA) was performed according to the manufacturer's instructions.

## Immunofluorescence

Frozen sections of the placentas were permeabilized with PBS containing 0.5% Triton X-100 (PBST) for 20 min and incubated for 1 hr with a blocking buffer in PBST. The sections were then incubated with anti-CD4 (sc-1176, Santa Cruz Biotechnology, China) and FITC-conjugated anti-human CD45RO (304204, BioLegend, USA) at 4°C overnight, followed by Alexa Fluor 568-conjugated secondary antibodies (dilution: 1:200, Yeasen, China) for 1 hr. The slides were counterstained with 4,6-diamidino-2-phenylindole (DAPI, 1 µg/ml; Roche, Switzerland) for 20 min. Digital images were obtained using confocal fluorescence microscopy (ZEISS LSM 800, Germany). ImageJ software was used to quantify the fluorescence intensity from immunofluorescence images.

## Statistical analysis

All experiments in the main text and figures consist of at least three biological replicates. We define biological replicates as the performance of the same experiment on different biological individuals, samples, or cell. Data were analyzed using SPSS version 20. After the Shapiro–Wilk test, the data were confirmed to be non-normal distribution. Kruskal–Wallis test was used to compare the results of experiments with multiple groups. Student's $t$-test was used to compare the results of experiments with two groups. All data are presented as mean ± SEM (*$p < 0.05$, **$p < 0.01$, ***$p < 0.001$; NS, not significant).

## Acknowledgements

The authors would like to express their heartfelt gratitude to the participants for their contributions.

## Additional information

### Funding

| Funder | Grant reference number | Author |
|---|---|---|
| National Natural Science Foundation of China | 82271694 | Lingling Jiang |
| Zhejiang Association for Science and Technology | LZ24H040002 | Lingling Jiang |
| National Natural Science Foundation of China | 82061160494 | Songying Zhang |
| Zhejiang Association for Science and Technology | 2023KY800 | Lingling Jiang |
| Zhejiang Association for Science and Technology | 2023C03033 | Dong Huang |

The funders had no role in study design, data collection and interpretation, or the decision to submit the work for publication.

### Author contributions

Haiyi Fei, Lingling Jiang, Conceptualization, Resources, Data curation, Software, Formal analysis, Supervision, Funding acquisition, Validation, Investigation, Visualization, Methodology, Writing – original draft, Project administration, Writing – review and editing; Xiaowen Lu, Investigation, Visualization, Methodology, Writing – original draft, Writing – review and editing; Zhan Shi, Formal analysis, Validation, Methodology, Writing – original draft, Writing – review and editing; Xiu Liu, Formal analysis, Investigation; Cuiyu Yang, Jianmin Wang, Resources, Supervision; Xiaohong Zhu, Validation, Visualization; Yuhan Lin, Resources, Methodology; Ziqun Jiang, Methodology, Writing – original draft; Dong Huang, Liu Liu, Resources, Supervision, Methodology; Songying Zhang, Conceptualization,

Resources, Supervision, Funding acquisition, Investigation, Writing – original draft, Writing – review and editing

### Author ORCIDs
Haiyi Fei (ID) http://orcid.org/0009-0002-3344-070X
Songying Zhang (ID) https://orcid.org/0000-0001-8044-6237
Lingling Jiang (ID) https://orcid.org/0000-0002-3343-6279

### Ethics

The clinical samples collected in this study have obtained informed consent from patients as well as consent for publication. The ethical approval guidelines that were followed Regulations on Ethical Review of the Biomedical Research Involving Humans (2016) issued by the National Health Commission of the People's Republic of China, the WMA Declaration of Helsinki, and the CIOMS International Ethical Guidelines for Biomedical Research. The ethical was approved by the Committee on the Medical Ethics Committee of the Sir Run Run Shaw Hospital of Zhejiang University (Permit Number: 20220207-30).

All of the animals were handled according to approved the Care and Use of Laboratory Animals (China). The protocol was approved by the Committee on the Animal Research Ethics Committee of the Sir Run Run Shaw Hospital of Zhejiang University (Permit Number: SRRSH202402608).

Reviewer #1 (Public review): https://doi.org/10.7554/eLife.100002.3.sa1
Reviewer #2 (Public review): https://doi.org/10.7554/eLife.100002.3.sa2
Author response https://doi.org/10.7554/eLife.100002.3.sa3

---

## Additional files

### Supplementary files
MDAR checklist

### Data availability

Sequencing data has been deposited in Dryad at DOI: 10.5061/dryad.4qrfj6qn0. All data generated or analyzed during this study is included in the manuscript and supporting files; source data files have been provided for Figures 1–7. Figure 1—source data, Figure 2—source data, Figure 3—source data, Figure 4—source data, Figure 5—source data, Figure 6—source data and, Figure 7—source data contain the numerical data used to generate the figures.

The following dataset was generated:

| Author(s) | Year | Dataset title | Dataset URL | Database and Identifier |
|---|---|---|---|---|
| Fei H, Lu X, Shi Z, Liu X, Yang C, Zhu X, Lin Y, Jiang Z, Wang J, Huang D, Liu L, Zhang S, Jiang L | 2025 | Data from: Deciphering the preeclampsia-specific immune microenvironment and the role of pro-inflammatory macrophages at the maternal-fetal interface | https://doi.org/10.5061/dryad.4qrfj6qn0 | Dryad Digital Repository, 10.5061/dryad.4qrfj6qn0 |

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
