## [Editor Report · eLife Assessment]

This **valuable** study investigates the immune system's role in pre-eclampsia. The authors map the immune cell landscape of the human placenta and find an increase in macrophages and Th17 cells in patients with pre-eclampsia. Following mouse studies, the authors suggest that the IGF1-IGF1R pathway might play a role in how macrophages influence T cells, potentially driving the pathology of pre-eclampsia. There is **convincing** evidence in this study that will be of interest to immunologists and developmental biologists.

---

## [Referee Report · Reviewer #1 (Public review)]

Summary:

This study explores the immune microenvironment of the placenta in preeclampsia (PE), which is often accompanied by gestational diabetes mellitus (GDM). Using CyTOF, they found that placentas from PE cases showed increased frequencies of memory-like Th17 cells, memory-like CD8⁺ T cells, and pro-inflammatory macrophages, alongside decreased levels of anti-inflammatory macrophages and granulocyte myeloid-derived suppressor cells (gMDSCs) compared to normal pregnancies. Further analysis revealed a positive correlation between pro-inflammatory macrophages and the expanded T cell populations, and a negative correlation with gMDSCs. Single-cell RNA sequencing provided mechanistic insights: transferring a specific subset of pro-inflammatory macrophages (F4/80⁺CD206⁻ with a distinct gene expression profile) from the uterus of PE mice to normal pregnant mice induced the formation of pathogenic memory-like Th17 cells via the IGF1-IGF1R pathway. This cellular interplay not only contributed to the development but also to the recurrence of PE. Additionally, these macrophages promoted the production of memory-like CD8⁺ T cells while inhibiting gMDSCs at the maternal-fetal interface, culminating in PE-like symptoms in mice. In conclusion, the study identifies a PE-specific immune cell network regulated by pro-inflammatory macrophages, offering new insights into the pathogenesis of preeclampsia.

Strengths:

Utilization of both human placental samples and multiple mouse models to explore the mechanisms linking inflammatory macrophages and T cells to preeclampsia (PE).

Incorporation of cutting-edge and complementary techniques such as CyTOF, scRNA-seq, bulk RNA-seq, and flow cytometry.

Identification of specific immune cell populations and their roles in PE.

Demonstration of the adverse effects of pro-inflammatory macrophages and T cells on pregnancy outcomes through in vivo manipulations.

Comments on revised version:

Several weaknesses were addressed during revision by conducting additional experiments, clarifying the manuscript's text, and incorporating new data that was not initially included.

---

## [Referee Report · Reviewer #2 (Public review)]

Summary:

Fei, Lu, Shi, et al. present a thorough evaluation of the immune cell landscape in pre-eclamptic human placentas by single-cell multi-omics methodologies compared to normal control placentas. Based on their findings of elevated frequencies of inflammatory macrophages and memory-like Th17 cells, they employ adoptive cell transfer mouse models to interrogate the coordination and function of these cell types in pre-eclampsia immunopathology. They demonstrate the putative role of the IGF1-IGF1R axis as the key pathway by which inflammatory macrophages in the placenta skew CD4+ T cells towards an inflammatory IL-17A-secreting phenotype that may drive tissue damage, vascular dysfunction, and elevated blood pressure in pre-eclampsia, leaving researchers with potential translational opportunities to pursue this pathway in this indication.

They present a major advance to the field in their profiling of human placental immune cells from pre-eclampsia patients where most extant single-cell atlases focus on term versus preterm placenta, or largely examine trophoblast biology with a much rarer subset of immune cells. While the authors present vast amounts of data at both the protein and RNA transcript level, we, the reviewers, feel this manuscript is still in need of much more clarity in its main messaging, and more discretion in including only key data that supports this main message most effectively.

Strengths:

(1) This study combines human and mouse analyses and allows for some amount of mechanistic insight into the role of pro-inflammatory and anti-inflammatory macrophages in the pathogenesis of pre-eclampsia (PE), and their interaction with Th17 cells.

(2) Importantly, they do this using matched cohorts across normal pregnancy and common PE comorbidities like gestation diabetes (GDM).

(3) The authors have developed clear translational opportunities from these "big data" studies by moving to pursue potential IGF1-based interventions.

[Editors' note: the authors have provided responses to the previously identified weaknesses]

---

## [Author Response]

The following is the authors’ response to the original reviews.

**Public Reviews:**

**Reviewer #1(Public review):**
Strengths:Utilization of both human placental samples and multiple mouse models to explore the mechanisms linking inflammatory macrophages and T cells to preeclampsia (PE).Incorporation of advanced techniques such as CyTOF, scRNA-seq, bulk RNA-seq, and flow cytometry.Identification of specific immune cell populations and their roles in PE, including the IGF1-IGF1R ligand-receptor pair in macrophage-mediated Th17 cell differentiation.Demonstration of the adverse effects of pro-inflammatory macrophages and T cells on pregnancy outcomes through transfer experiments.Weaknesses:Comment 1. Inconsistent use of uterine and placental cells, which are distinct tissues with different macrophage populations, potentially confounding results.

Response1: We thank the reviewers' comments. We have done the green fluorescent protein (GFP) pregnant mice-related animal experiment, which was not shown in this manuscript. The wild-type (WT) female mice were mated with either transgenic male mice, genetically modified to express GFP, or with WT male mice, in order to generate either GFP-expressing pups (GFP-pups) or their genetically unmodified counterparts (WT-pups), respectively. Mice were euthanized on day 18.5 of gestation, and the uteri of the pregnant females and the placentas of the offspring were analyzed using flow cytometry. The majority of macrophages in the uterus and placenta are of maternal origin, which was defined by GFP negative. In contrast, fetal-derived macrophages, distinguished by their expression of GFP, represent a mere fraction of the total macrophage population. We have added the GFP pregnant mice-related data in uterine and placental cells (Line204-212).

Comment 2. Missing observational data for the initial experiment transferring RUPP-derived macrophages to normal pregnant mice.

Response 2: We thank the reviewers' comments. We have added the observational data (Figure 4-figure supplement 1D, 1E)and a corresponding description of the data (Line 198-203).

Comment 3. Unclear mechanisms of anti-macrophage compounds and their effects on placental/fetal macrophages.

Response 3: We thank the reviewers' comments. PLX3397, the inhibitor of CSF1R, which is needed for macrophage development (Nature. 2023, PMID: 36890231; Cell Mol Immunol. 2022, PMID: 36220994), we have stated that on Line 227-230. However, PLX3397 is a small molecule compound that possesses the potential to cross the placental barrier and affect fetal macrophages. We have discussed the impact of this factor on the experiment in the Discussion section (Line457-459).

Comment 4. Difficulty in distinguishing donor cells from recipient cells in murine single-cell data complicates interpretation.

Response 4: We thank the reviewers' comments. Upon analysis, we observed a notable elevation in the frequency of total macrophages within the CD45^+^ cell population. Then we subsequently performed macrophage clustering and uncovered a marked increase in the frequency of Cluster 0, implying a potential correlation between Cluster 0 and donor-derived cells. RNA sequencing revealed that the F480^+^CD206^-^ pro-inflammatory donor macrophages exhibited a Folr2^+^Ccl7^+^Ccl8^+^C1qa^+^C1qb^+^C1qc^+^ phenotype, which is consistent with the phenotype of cluster 0 in macrophages observed in single-cell RNA sequencing (Figure 4D and Figure 5E). Therefore, we believe that the donor cells should be cluster 0 in macrophages.

Comment 5. Limitation of using the LPS model in the final experiments, as it more closely resembles systemic inflammation seen in endotoxemia rather than the specific pathology of PE.

Response 5: We thank the reviewers' comments. Firstly, our other animal experiments in this manuscript used the Reduction in Uterine Perfusion Pressure (RUPP) mouse model to simulate the pathology of PE. However, the RUPP model requires ligation of the uterine arteries in pregnant mice on day 12.5 of gestation, which hinders T cells returning from the tail vein from reaching the maternal-fetal interface. In addition, this experiment aims to prove that CD4^+^ T cells are differentiated into memory-like Th17 cells through IGF-1R receptor signaling to affect pregnancy by clearing CD4^+^ T cells in vivo with an anti-CD4 antibody followed by injecting IGF-1R inhibitor-treated CD4^+^ T cells. And we proved that injection of RUPP-derived memory-like CD4^+^ T cells into pregnant mice induces PE-like symptoms (Figure 6F-6H). In summary, the application of the LPS model in the final experiments does not affect the conclusions.

**Reviewer #2 (Public review):**
Strengths:(1) This study combines human and mouse analyses and allows for some amount of mechanistic insight into the role of pro-inflammatory and anti-inflammatory macrophages in the pathogenesis of pre-eclampsia (PE), and their interaction with Th17 cells.(2) Importantly, they do this using matched cohorts across normal pregnancy and common PE comorbidities like gestation diabetes (GDM).(3) The authors have developed clear translational opportunities from these "big data" studies by moving to pursue potential IGF1-based interventions.Weaknesses:(1) Clearly the authors generated vast amounts of multi-omic data using CyTOF and single-cell RNA-seq (scRNA-seq), but their central message becomes muddled very quickly. The reader has to do a lot of work to follow the authors' multiple lines of inquiry rather than smoothly following along with their unified rationale. The title description tells fairly little about the substance of the study. The manuscript is very challenging to follow. The paper would benefit from substantial reorganizations and editing for grammatical and spelling errors. For example, RUPP is introduced in Figure 4 but in the text not defined or even talked about what it is until Figure 6. (The figure comparing pro- and anti-inflammatory macrophages does not add much to the manuscript as this is an expected finding).

Response 1: We thank the reviewers' comments. According to the reviewer's suggestion, we have made the necessary revisions. Firstly, the title of the article has been modified to be more specific. We also introduce the RUPP mouse model when interpreted Figure 4-figure supplement 1. Thirdly, We have moved the images of Figure 7 to the Figure 6-figure supplement 2 make them easier to follow. Finally, we diligently corrected the grammatical and spelling errors in the article. As for the figure comparing pro- and anti-inflammatory macrophages, the Editor requested a more comprehensive description of the macrophage phenotype during the initial submission. As a result, we conducted the transcriptome RNA-seq of both uterine-derived pro-inflammatory and anti-inflammatory macrophages and conducted a detailed analysis of macrophages in scRNA-seq.

Comment 2. The methods lack critical detail about how human placenta samples were processed. The maternal-fetal interface is a highly heterogeneous tissue environment and care must be taken to ensure proper focus on maternal or fetal cells of origin. Lacking this detail in the present manuscript, there are many unanswered questions about the nature of the immune cells analyzed. It is impossible to figure out which part of the placental unit is analyzed for the human or mouse data. Is this the decidua, the placental villi, or the fetal membranes? This is of key importance to the central findings of the manuscript as the immune makeup of these compartments is very different. Or is this analyzed as the entirety of the placenta, which would be a mix of these compartments and significantly less exciting?

Response 2: We thank the reviewers' comments. Placental villi rather than fetal membranes and decidua were used for CyToF in this study. This detail about how human placenta samples were processed have been added to the Materials and Methods section (Line564-576).

Comment 3. Similarly, methods lack any detail about the analysis of the CyTOF and scRNAseq data, much more detail needs to be added here. How were these clustered, what was the QC for scRNAseq data, etc? The two small paragraphs lack any detail.

Response 3: We thank the reviewers' comments. The details about the analysis of the CyTOF (Line577-586) and scRNAseq (Line600-615) data have been added in the Materials and Methods section.

Comment 4. There is also insufficient detail presented about the quantities or proportions of various cell populations. For example, gdT cells represent very small proportions of the CyTOF plots shown in Figures 1B, 1C, & 1E, yet in Figures 2I, 2K, & 2K there are many gdT cells shown in subcluster analysis without a description of how many cells are actually represented, and where they came from. How were biological replicates normalized for fair statistical comparison between groups?

Response 4: We thank the reviewers' comments. In our study, approximately 8×10^^5^ cells were collected per group for analysis using CyTOF. Of these, about 10% (8×10^^4^ cells per group) were utilized to generate Figure 1B. As depicted in Figure 1B, gdT cells constitute roughly 1% of each group, with specific percentages as follows: NP group (1.23%), PE group (0.97%), GDM group (0.94%), and GDM&PE group (1.26%), which equates to approximately 800 cells per group. For the subsequent gdT cell analysis presented in Figure 2I, we employed data from all cells within each group to construct the tSNE maps, comprising approximately 8000 cells per group. Consequently, it may initially appear that the number of gdT cells is significantly higher than what is shown in Figure 1B. To clarify this, we have included pertinent explanations in the figure legend. Given the relatively low proportions of gdT cells, we did not pursue further investigations of these cells in subsequent experiments. Following your suggestion, we have relocated this result to the supplementary materials, where it is now presented as Figure 2-figure supplement 1D-E.

The number of biological replicates (samples) is consistent with Figure 1, and this information has been added to the figure legend.

Comment 5. The figures themselves are very tricky to follow. The clusters are numbered rather than identified by what the authors think they are, the numbers are so small, that they are challenging to read. The paper would be significantly improved if the clusters were clearly labeled and identified. All the heatmaps and the abundance of clusters should be in separate supplementary figures.

Response 5: We thank the reviewers' comments. Based on your suggestions, we have labeled and defined the Clusters (Figure 2A, 2F, Figure 3A, Figure 5C and Figure 6A). Additionally, we have moved most of the heatmaps to the supplementary materials.

Comment 6. The authors should take additional care when constructing figures that their biological replicates (and all replicates) are accurately represented. Figure 2H-2K shows N=10 data points for the normal pregnant (NP) samples when clearly their Table 1 and test denote they only studied N=9 normal subjects.

Response 6: We thank the reviewers' careful checking. During our verification, we found that one sample in the NP group had pregnancy complications other than PE and GDM. The data in Figure 2H-2K was not updated in a timely manner. We have promptly updated this data and reanalyze it.

Comment 7. There is little to no evaluation of regulatory T cells (Tregs) which are well known to undergird maternal tolerance of the fetus, and which are well known to have overlapping developmental trajectory with RORgt+ Th17 cells. We recommend the authors evaluate whether the loss of Treg function, quantity, or quality leaves CD4+ effector T cells more unrestrained in their effect on PE phenotypes. References should include, accordingly: PMCID: PMC6448013 / DOI: 10.3389/fimmu.2019.00478; PMC4700932 / DOI: 10.1126/science.aaa9420.

Response 7: We thank the reviewers' comments. We have done the Treg-related animal experiment, which was not shown in this manuscript. We have added the Treg-related data in Figure 6F-6H. The injection of CD4^+^CD44^+^ T cells derived from RUPP mouse, characterized by a reduced frequency of Tregs, could induce PE-like symptoms in pregnant mice (Line297-304). Additionally, we have added a necessary discussion about Tregs and cited the literature you mentioned (Line433-439).

Comment 8. In discussing gMDSCs in Figure 3, the authors have missed key opportunities to evaluate bona fide Neutrophils. We recommend they conduct FACS or CyTOF staining including CD66b if they have additional tissues or cells available. Please refer to this helpful review article that highlights key points of distinguishing human MDSC from neutrophils: https://doi.org/10.1038/s41577-024-01062-0. This will both help the evaluation of potentially regulatory myeloid cells that may suppress effector T cells as well as aid in understanding at the end of the study if IL-17 produced by CD4+ Th17 cells might recruit neutrophils to the placenta and cause ROS immunopathology and fetal resorption.

Response 8: We thank the reviewers' comments. Although we do not have additional tissues or cells available to conduct FACS or CyTOF staining, including for CD66b, we have utilized CD15 and CD66b antibodies for immunofluorescence stain of placental tissue, and our findings revealed a pronounced increase in the proportion of neutrophils among PE patients, fostering the hypothesis that IL-17A produced by Th17 cells might orchestrate the migration of neutrophils towards the placental milieu (Figure 6-figure supplement 2F; Line 325-328). We have cited these references and discussed them in the Discussion section (Line 459-465).

Comment 9. Depletion of macrophages using several different methodologies (PLX3397, or clodronate liposomes) should be accompanied by supplementary data showing the efficiency of depletion, especially within tissue compartments of interest (uterine horns, placenta). The clodronate piece is not at all discussed in the main text. Both should be addressed in much more detail.

Response 9: We thank the reviewers' comments. We already have the additional data on the efficiency of macrophage depletion involving PLX3397 and clodronate liposomes, which were not present in this manuscript, and we'll add it to the Figure 4-figure supplement 2A,2B. The clodronate piece is mentioned in the main text (Line236-239), but only briefly described, because the results using clodronate we obtained were similar to those using PLX3397.

Comment 10. There are many heatmaps and tSNE / UMAP plots with unhelpful labels and no statistical tests applied. Many of these plots (e.g. Figure 7) could be moved to supplemental figures or pared down and combined with existing main figures to help the authors streamline and unify their message.

Response 10: We thank the reviewers' comments. We have moved the images of Figure 7 to the Figure 6-figure supplement 2. We also have moved most of the heatmaps to the supplementary materials.

Comment 11. There are claims that this study fills a gap that "only one report has provided an overall analysis of immune cells in the human placental villi in the presence and absence of spontaneous labor at term by scRNA-seq (Miller 2022)" (lines 362-364), yet this study itself does not exhaustively study all immune cell subsets...that's a monumental task, even with the two multi-omic methods used in this paper. There are several other datasets that have performed similar analyses and should be referenced.

Response 11: We thank the reviewers' comments. We have search for more literature and reference additional studies that have conducted similar analyses (Line382-393).

Comment 12. Inappropriate statistical tests are used in many of the analyses. Figures 1-2 use the Shapiro-Wilk test, which is a test of "goodness of fit", to compare unpaired groups. A Kruskal-Wallis or other nonparametric t-test is much more appropriate. In other instances, there is no mention of statistical tests (Figures 6-7) at all. Appropriate tests should be added throughout.

Response 12: We thank the reviewers' comments. As stated in the Statistical Analysis section (lines 672-676), the Kruskal-Wallis test was used to compare the results of experiments with multiple groups. Comparisons between the two groups in Figures 5 were conducted using Student's t-test. The aforementioned statistical methods have been included in the figure legends.

**Recommendations for the authors:**

**Reviewer #1 (Recommendations for the authors):**
Overall, the study has several strengths, including the use of human samples and animal models, as well as the incorporation of multiple cutting-edge techniques. However, there are some significant issues with the murine model experiments that need to be addressed:Comment 1. The authors are not consistent in their use of or focus on uterine and placental cells. These are distinct tissues, and numerous prior reports have indicated differences in the macrophage populations of these tissues, due in part to the predominantly maternal origin of macrophages in the uterus and the largely fetal origin of those in the placenta. The rationale for switching between uterine and placental cells in different experiments is not clear, and the inclusion of cells from both (such as in the bulk RNAseq experiments) could be potentially confounding.

Response 1: We thank the reviewers' comments. We have done the green fluorescent protein (GFP) pregnant mice-related animal experiment, which was not shown in this manuscript. The wild-type (WT) female mice were mated with either transgenic male mice, genetically modified to express GFP, or with WT male mice, in order to generate either GFP-expressing pups (GFP-pups) or their genetically unmodified counterparts (WT-pups), respectively. Mice were euthanized on day 18.5 of gestation, and the uteri of the pregnant females and the placentas of the offspring were analyzed using flow cytometry. The majority of macrophages in the uterus and placenta are of maternal origin, which was defined by GFP negative. In contrast, fetal-derived macrophages, distinguished by their expression of GFP, represent a mere fraction of the total macrophage population, signifying their inconsequential or restricted presence amidst the broader cellular landscape. We have added the GPF pregnant mice-related data in Figure 4-figure supplement 1D-1E to explain the different macrophage populations in the uterine and placental cells.

Comment 2. The observational data for the initial experiment transferring RUPP-derived macrophages to normal pregnant mice (without any other manipulations) seems to be missing. They do not seem to be presented in Figure 4 where they are expected based on the results text.

Response 2: We thank the reviewers' comments. We thank the reviewers' comments. We have added the observational data (Figure 4-figure supplement 1D, 1E) and a corresponding description of the data (Line 198-203).

Comment 3. The action of the anti-macrophage compounds is not well explained, nor are their mechanisms validated as affecting or not affecting the placental/fetal macrophage populations. It is important to clarify whether the macrophages are depleted or merely inhibited by these treatments, and it is absolutely critical to determine whether these treatments are affecting placental/fetal macrophage populations (the latter indicative of placental transfer), given the focus on placental macrophages.

Response 3: We thank the reviewers' comments. PLX3397, the inhibitor of CSF1R, which is needed for macrophage development (Nature. 2023, PMID: 36890231; Cell Mol Immunol. 2022, PMID: 36220994), we have stated that on Line227-230. However, PLX3397 is a small molecule compound that possesses the potential to cross the placental barrier and affect fetal macrophages. We will discuss the impact of this factor on the experiment in the Discussion section (Line457-459).

Comment 4. The interpretation of the murine single-cell data is hampered by the lack of means for distinguishing donor cells from recipient cells, which is important when seeking to identify the influence of the donor cells.

Response 4: We thank the reviewers' comments. Upon analysis, we observed a notable elevation in the frequency of total macrophages within the CD45^+^ cell population. Then we subsequently per formed macrophage clustering and uncovered a marked increase in the frequency of Cluster 0, implying a potential correlation between Cluster 0 and donor-derived cells. RNA sequencing revealed that the F480^+^CD206^-^ pro-inflammatory donor macrophages exhibited a Folr2^+^Ccl7^+^Ccl8^+^C1qa^+^C1qb^+^C1qc^+^ phenotype, which is consistent with the phenotype of cluster 0 in macrophages observed in single-cell RNA sequencing (Figure 4D and Figure 5E). Therefore, the donor cells should be in cluster 0 in macrophages.

Comment 5. The switch to the LPS model in the final experiments is a limitation, as this model more closely resembles the systemic inflammation seen in endotoxemia rather than the specific pathology of preeclampsia (PE). While this is not an exhaustive list, the number of weaknesses in the experimental design makes it difficult to evaluate the findings comprehensively.

Response 5: We thank the reviewers' comments. Firstly, our other animal experiments in this manuscript used the RUPP mouse model to simulate the pathology of PE. However, the RUPP model requires ligation of the uterine arteries in pregnant mice on day 12.5 of gestation, which hinders T cells returning from the tail vein from reaching the maternal-fetal interface. In addition, this experiment aims to prove that CD4^+^ T cells are differentiated into memory-like Th17 cells through IGF-1R receptor signaling to affect pregnancy by clearing CD4^+^ T cells in vivo with an anti-CD4 antibody followed by injecting IGF-1R inhibitor-treated CD4^+^ T cells. We proved that injection of RUPP-derived memory-like CD4^+^ T cells into pregnant rats induces PE-like symptoms (Figure 6F-6H). In summary, applying the LPS model in the final experiments does not affect the conclusions.

Minor comments:Comment 1. Introduction, Lines 67-74: The phrasing here is unclear as to the roles that each mentioned immune cell subset is playing in preeclampsia. Given the statement "Elevated levels of maternal inflammation...", does this imply that the numbers of all mentioned immune cell subsets are increased in the maternal circulation? If not, please consider rewording this.

Response 1: We thank the reviewers' comments. We have revised the manuscript as follows: Currently, the pivotal mechanism underpinning the pathogenesis of preeclampsia is widely acknowledged to involve an increased frequency of pro-inflammatory M1-like maternal macrophages, along with an elevation in Granulocytes capable of superoxide generation, CD56^+^ CD94^+^ natural killer (NK) cells, CD19^+^CD5^+^ B1 lymphocytes, and activated γδ T cells. Conversely, this pathological process is accompanied by a notable decrease in the frequency of anti-inflammatory M2-like macrophages and NKp46^+^ NK cells (Line67-77).

Comment 2. Introduction, Lines 67-80: Is the involvement of the described immune cell subsets largely ubiquitous to preeclampsia? Recent multi-omic studies suggest that preeclampsia is a heterogeneous condition with different subsets, some more biased towards systemic immune activation than others. Thus, it is important to clarify whether the involvement of specific immune subsets is generally observed or more specific.

Response 2: We thank the reviewers' comments. We have added a new paragraph as follows: Moreover, as PE can be subdivided into early- and late-onset PE diagnosed before 34 weeks or from 34 weeks of gestation, respectively. Research has revealed that among the myriad of cellular alterations in PE, pro-inflammatory M1-like macrophages and intrauterine B1 cells display an augmented presence at the maternal-fetal interface of both early-onset and late-onset PE patients. Decidual natural killer (dNK) cells and neutrophils emerge as paramount contributors, playing a more crucial role in the pathogenesis of early-onset PE than late-onset PE (Front Immunol. 2020. PMID: 33013837) (Line83-89).

Comment 3. Introduction, Lines 81-86: The point of this short paragraph is not clear; the authors mention two very specific cellular interactions without explaining why.

Response 3: In the previous paragraph, we uncovered a heightened inflammatory response among multiple immune cells in patients with PE, yet the intricate interplay between these individual immune cells has been seldom elucidated in the context of PE patient. This is precisely why we delve into the realm of specific immune cellular interactions in relation to other pregnancy complications in this paragraph (Line91-98).

Comment 4. Methods: What placental tissues (e.g., villous tree, chorionic plate, extraplacental membranes) were included for CyTOF analysis? Was any decidual tissue (e.g., basal plate) included? Please clarify.

Response 4: Placental villi rather than chorionic plate and extraplacental membranes were used for CyToF in this study. The relevant content has been incorporated into the "Materials and Methods" section (Line564-576).

Comment 5. Results, Table 1: The authors should clarify that all PE samples were not full term (i.e., were less than 37 weeks of gestation), which is to be expected. In addition, were the PE cases all late-onset PE?

Response 5: All PE samples enumerated in Table 1 demonstrate a late-onset preeclampsia, with placental specimens being procured from patients more than 35 weeks of gestation and less than the 38 weeks of pregnancy. The relevant content has been incorporated into the "Materials and Methods" section (Line574-576).

Comment 6. Results, Figure 1: Are the authors considering the identified Macrophage cluster as being largely fetal (e.g., Hofbauer cells)? This also depends on whether any decidual tissue was included in the placental samples for CyTOF.

Response 6: Firstly, the specimens subjected to CyToF analysis were devoid of decidual tissue and exclusively comprised placental villi. Secondly, the Macrophage cluster in Figure 1 undeniably encompasses Hofbauer cells, and we considering fetal-derived macrophages likely constituting the substantial proportion of the cellular population. However, a limitation of the CyToF technique lies in its inability to discern between maternal and fetal origins of these cells, thereby precluding a definitive distinction.

Comment 7. Results, Figure 2C: Did the authors validate other T-cell subset markers (e.g., Th1, Th2, Th9, etc.)?

Response 7: In this study, we did not validate additional T-cell subset markers presented in Figure 2C, recognizing the potential for deeper insights. As we embark on our subsequent research endeavors, we aim to meticulously explore and characterize the intricate changes in diverse T-cell populations at the maternal-fetal interface, with a particular focus on preeclampsia patients, thereby advancing our understanding of this complex condition.

Comment 8. Results, Figure 2D: Where were the detected memory-like T cells located in the placenta? Did they cluster in certain areas or were they widely distributed?

Response 8: Upon a thorough re-evaluation of the immunofluorescence images specific to the placenta, we observed a notable preponderance of memory-like T cells residing within the placental sinusoids (Line135-139).

Comment 9. Results, Figure 2E: I would suggest separating the two plots so that the Y-axis can be expanded for TIM3, as it is impossible to view the medians currently.

Response 9: We thank the reviewers' comments. We have made the adjustment to Figure 2E according to the reviewers' suggestions.

Comment 10. Results, Lines 138-140: Do the authors consider that the altered T-cells are largely resident cells of the placenta or newly invading/recruited cells? The clarification of distribution within the placental tissues as mentioned above would help answer this.

Response 10: Our analysis revealed the presence of memory-like T cells within the placental sinusoids, as evident from the immunofluorescence examination of placental tissues. Consequently, these T cells may represent recently recruited cellular entities, traversing the placental vasculature and integrating into this unique maternal-fetal microenvironment (Line135-139).

Comment 11. Results, Figure 3C: Has a reduction of gMDSCs (or MDSCs in general) been previously reported in PE?

Response 11: Myeloid-derived suppressor cells (MDSCs) constitute a diverse population of myeloid-derived cells that exhibit immunosuppressive functions under various conditions. Previous reports have documented a decrease in the levels of gMDSCs from peripheral blood or umbilical cord blood among patients with preeclampsia (Am J Reprod Immunol. 2020, PMID: 32418253; J Reprod Immunol. 2018, PMID: 29763854; Biol Reprod. 2023, PMID: 36504233). Nevertheless, there was no documented reports thus far on the alterations and specific characteristics in gMDSCs within the placenta of PE patients.

Comment 12. Results, Figure 3D-E: It is not clear what new information is added by the correlations, as the increase of both cluster 23 in CD11b+ cells and cluster 8 in CD4+ T cells in PE cases was already apparent. Are these simply to confirm what was shown from the quantification data?

Response 12: Despite the evident increase in both cluster 23 within CD11b^+^ cells and cluster 8 within CD4^+^ T cells in PE cases, the existence of a potential correlation between these two clusters remains elusive. To gain insight into this question, we conducted a Pearson correlation analysis, which is presented in Figure 3D-E, revealing a positive correlation between the two clusters.

Comment 13. Results, Figure 4A: Please clarify in the results text that the RNA-seq of macrophages from RUPP mice was performed prior to their injection into normal pregnant mice.

Response 13: We thank the reviewers' comments. We have updated Figure 4A according to the reviewers' suggestions.

Comment 14. Results / Methods, Figure 4: For the transfer of macrophages from RUPP mice into normal mice, why were the uterine tissues included to isolate cells? The uterine macrophages will be almost completely maternal, as opposed to the largely fetal placental macrophages, and despite the sorting for specific markers these are likely distinct subsets that have been combined for injection. This could potentially impact the differential gene expression analysis and should be accounted for. In addition, did murine placental samples include decidua? This should be clarified.

Response 14: We thank the reviewers' comments. For our experimental design involving human samples, we meticulously selected placental tissue as the primary focus. Initially, we aimed for uniformity by contemplating the utilization of mouse placenta. However, a pivotal revelation emerged from the GFP pregnant mice-related data in Figure 4-figure supplement 1D,1E: the uterus and placenta of mice are predominantly populated by maternal macrophages, with fetal macrophages virtually absent, marking a notable divergence from the human scenario. Furthermore, the uterine milieu exhibits a macrophage concentration exceeding 20% of total cellular composition, whereas in the placenta, this proportion dwindles to less than 5%, underscoring a distinct distribution pattern. Given these discrepancies and considerations, we incorporated mouse uterine tissues into our protocol to isolate cells, ensuring a more comprehensive and informative exploration that acknowledges the inherent differences between human and mouse placental biology.

Comment 15. Results, Lines 186-187: I think the figure citation should be Figure 4D here.

Response 15: We thank the reviewers' careful checking. We have revised and updated Figure 4 accordingly.

Comment 16. Results, Figure 4: Where are the results of the injection of anti-inflammatory and pro-inflammatory macrophages into normal mice? This experiment is mentioned in Figure 4A, but the only results shown in Figure 4 are with the PLX3397 depletion.

Response 16: The aim of this experiment in figure 4 is to conclusively ascertain the influence of pro-inflammatory and anti-inflammatory macrophages on the other immune cells within the maternal-fetal interface, as well as their implications for pregnancy outcomes. To achieve this, we employed a strategic approach involving the administration of PLX3397, a compound capable of eliminating the preexisting macrophages in mice. Subsequently, anti-inflam or pro-inflam macrophages were injected to these mice, thereby eliminating the confounding influence of the native macrophage population. This methodology allows for a more discernible observation of the specific effects these two types of macrophages exert on the immune landscape at the maternal-fetal interface and their ultimate impact on pregnancy outcomes.

Comment 17. Results, Lines 189-190: Does PLX3397 inhibit macrophage development/signaling/etc. or result in macrophage depletion? This is an important distinction. If depletion is induced, does this affect placental/fetal macrophages or just maternal macrophages?

Response 17: We thank the reviewers' comments. We have updated the additional data on the efficiency of macrophage depletion involving PLX3397 in Figure 4-figure supplement 2A. PLX3397 is a small molecule compound that possesses the potential to cross the placental barrier and affect fetal macrophages. We have discussed the impact of this factor on the experiment in the Discussion section (Line457-459).

Comment 18. Results, Lines 197-198: Similarly, does clodronate liposome administration affect only maternal macrophages, or also placental/fetal macrophages?

Response 18: We thank the reviewers' comments. We have updated the additional data on the efficiency of macrophage depletion involving Clodronate Liposomes in Figure 4-figure supplement 2B. Clodronate Liposomes, which are intricate vesicles encapsulating diverse substances, while only small molecule compounds possess the potential to cross the placental barrier. Consequently, we hold the view that the influence of these liposomes is likely confined to the maternal macrophages (Artif Cells Nanomed Biotechnol. 2023. PMID: 37594208).

Comment 19. Results, Line 206: A minor point, but consider continuing to refer to the preeclampsia model mice as RUPP mice rather than PE mice.

Response 19: We thank the reviewers' comments. We have revised and updated this section accordingly.

Comment 20. Results / Methods, Figure 5: For these experiments, why did the authors focus on the mouse uterus?

Response 20: We have previously addressed this query in our Response 14. We incorporated mouse uterine tissues for cell isolation due to the profound differences in placental biology between humans and mice.

Comment 21. Results, Figure 5: Did the authors have a means of distinguishing the transferred donor cells from the recipient cells for their single-cell analysis? If the goal is to separate the effects of the macrophage transfer on other uterine immune cells, then it would be important to identify and separate the donor cells.

Response 21: We thank the reviewers' comments. Upon analysis, we observed a notable elevation in the frequency of total macrophages within the CD45^+^ cell population. Then we subsequently performed macrophage clustering and uncovered a marked increase in the frequency of Cluster 0, implying a potential correlation between Cluster 0 and donor-derived cells. RNA sequencing revealed that the F480^+^CD206^-^ pro-inflammatory donor macrophages exhibited a Folr2^+^Ccl7^+^Ccl8^+^C1qa^+^C1qb^+^C1qc^+^ phenotype, which is consistent with the phenotype of cluster 0 in macrophages observed in single-cell RNA sequencing (Figure 4D and Figure 5E). Therefore, the donor cells should be in cluster 0 in macrophages.

Comment 22. Results, Lines 247-248: While the authors have prudently noted that the observed T-cell phenotypes are merely suggestive of immunosuppression, any claims regarding changes in the immunosuppressive function after macrophage transfer would require functional studies of the T cells.

Response 22: We thank the reviewers' comments. Upon revisiting and meticulously reviewing the pertinent literature, we have refined our terminology, transitioning from 'immunosuppression' to 'immunomodulation', thereby enhancing the accuracy and precision of our Results (Line285-287).

Comment 23. Results, Figure 6G: The observation of worsened outcomes and PE-like symptoms after T-cell transfer is interesting, but other models of PE induced by the administration of Th1-like cells have already been reported. Are the authors' findings consistent with these reports? These findings are strengthened by the evaluation of second-pregnancy outcomes following the transfer of T cells in the first pregnancy.

Response 23: We thank the reviewers' comments. As we verified in Figure 6F-6H, the injection of CD4^+^CD44^+^ T cells derived from RUPP mouse, characterized by a reduced frequency of Tregs and an increased frequency of Th17 cells, could induce PE-like symptoms in pregnant mice. In line with other studies, which have implicated Th1-like cells in the manifestation of PE-like symptoms, we posit a novel hypothesis: beyond Th1 cells, Th17 cells also have the potential to induce PE-like symptoms.

Comment 24. Results, Lines 327-337: The disease model implied by the authors here is not clear. Given that the authors' human findings are in the placental macrophages, are the authors proposing that placental macrophages are induced to an M1 phenotype by placenta-derived EVs? Please elaborate on and clarify the proposed model.

Response 24 In the article authored by our team, titled "Trophoblast-Derived Extracellular Vesicles Promote Preeclampsia by Regulating Macrophage Polarization" published in Hypertension (Hypertension. 2022, PMID: 35993233), we employed trophoblast-derived extracellular vesicles isolated from PE patients as a means to induce an M1-like macrophage phenotype in macrophages from human peripheral blood in vitro. Consequently, in the present study, we have directly leveraged this established methodology to induce pro-inflammatory macrophages.

Comment 25. Results / Methods, Figure 8E-H: What is the reasoning for switching to an LPS model in this experiment? LPS is less specific to PE than the RUPP model.

Response 25: We thank the reviewers' comments. Firstly, our other animal experiments in this manuscript used the RUPP mouse model to simulate the pathology of PE. However, the RUPP model requires ligation of the uterine arteries in pregnant mice on day 12.5 of gestation, which hinders T cells returning from the tail vein from reaching the maternal-fetal interface. In addition, this experiment aims to prove that CD4^+^ T cells are differentiated into memory-like Th17 cells through IGF-1R receptor signaling to affect pregnancy by clearing CD4^+^ T cells in vivo with an anti-CD4 antibody followed by injecting IGF-1R inhibitor-treated CD4^+^ T cells. And we proved that injection of RUPP-derived memory-like CD4^+^ T cells into pregnant mice induces PE-like symptoms (Figure 6). In summary, the application of the LPS model in the final experiments does not affect the conclusions.

Comment 26. Discussion: What do the authors consider to be the origins of the inflammatory cells associated with PE onset? Are these maternal cells invading the placental tissues, or are these placental resident (likely fetal) cells?

Response 26: We thank the reviewers' comments. Numerous reports have consistently observed the presence of inflammatory cells and factors in the maternal peripheral blood and placenta tissues of PE patients, fostering the prevailing notion that the progression of PE is intricately linked to the maternal immune system's inflammatory response towards the fetus. Nevertheless, intriguing findings from single-cell RNA sequencing, analyzed through bioinformatic methods, have challenged this perspective (Elife. 2019. PMID: 31829938；Proc Natl Acad Sci U S A. 2017.PMID: 28830992). These studies reveal that the placenta harbors not just immune cells of maternal origin but also those of fetal origin, raising questions about whether these are maternal cells infiltrating placental tissues or resident (possibly fetal) placental cells. Further investigation is imperative to elucidate this complex interplay.

Comment 27. Discussion: Given the observed lack of changes in the GDM or GDM+PE groups, do the authors consider that GDM represents a distinct pathology that can lead to secondary PE, and thus is different from primary PE without GDM?

Response 27: It's possible. Though previous studies reported GDM is associated with aberrant maternal immune cell adaption the findings remained controversial. It seems that GDM does not induce significant alterations in placental immune cell profile in our study, which made us pay more attention to the immune mechanism in PE. However, it is confusing for the reasons why individuals with GDM&PE were protected from the immune alterations at the maternal fetal interface. Limited placental samples in the GDM&PE group can partly explain it, for it is hard to collect clean samples excluding confounding factors. A study reported that macrophages in human placenta maintained anti-inflammatory properties despite GDM (Front Immunol, 2017, PMID: 28824621).Barke et al. also found that more CD163^+^ cells were observed in GDM placentas compared to normal controls (PLoS One, 2014, PMID: 24983948). Thus, GDM is likely to have a protective property in the placental immune environment when the individuals are complicated with PE.

**Reviewer #2 (Recommendations for the authors):**
Comment 1. IF images need to be quantified.

Response 1: We thank the reviewers' comments. We have quantified and calculated the fluorescence intensity and added it in Figure 2D.

Comment 2. Cluster 12 in Figure 3 is labeled as granulocytes but listed under macrophages.

Response 2: We thank the reviewers' careful checking. We have revised and updated Figure 3A.

Comment 3. Figure 4 labels in the text and figure do not match, no 4G in the figure.

Response 3: We thank the reviewers' careful checking. The figure labels of Figure 4 have been revised and updated.